# Temperature-dependent interphase formation and Li⁺ transport in lithium metal batteries

Suting Weng ⬡[1,2], Xiao Zhang[1,3], Gaojing Yang ⬡[1,2], Simeng Zhang[1,3], Bingyun Ma[4], Qiuyan Liu[1,3], Yue Liu[4], Chengxin Peng[5], Huixin Chen ⬡[6], Hailong Yu[1], Xiulin Fan ⬡[7], Tao Cheng ⬡[4], Liquan Chen[1], Yejing Li ⬡[1] ✉, Zhaoxiang Wang[1,2,3] ✉ & Xuefeng Wang ⬡[1,2,3,8] ✉

High-performance Li-ion/metal batteries working at a low temperature (i.e., <−20 °C) are desired but hindered by the sluggish kinetics associated with Li⁺ transport and charge transfer. Herein, the temperature-dependent Li⁺ behavior during Li plating is profiled by various characterization techniques, suggesting that Li⁺ diffusion through the solid electrolyte interface (SEI) layer is the key rate-determining step. Lowering the temperature not only slows down Li⁺ transport, but also alters the thermodynamic reaction of electrolyte decomposition, resulting in different reaction pathways and forming an SEI layer consisting of intermediate products rich in organic species. Such an SEI layer is metastable and unsuitable for efficient Li⁺ transport. By tuning the solvation structure of the electrolyte with a lower lowest unoccupied molecular orbital (LUMO) energy level and polar groups, such as fluorinated electrolytes like 1 mol L⁻¹ lithium bis(fluorosulfonyl)imide (LiFSI) in methyl trifluoroacetate (MTFA): fluoroethylene carbonate (FEC) (8:2, weight ratio), an inorganic-rich SEI layer more readily forms, which exhibits enhanced tolerance to a change of working temperature (thermodynamics) and improved Li⁺ transport (kinetics). Our findings uncover the kinetic bottleneck for Li⁺ transport at low temperature and provide directions to enhance the reaction kinetics/thermodynamics and low-temperature performance by constructing inorganic-rich interphases.

Lithium-ion batteries (LIBs) operating at a low temperature are highly wanted in the cold seasons or locations for different applications such as electric vehicles, submarines, and airplanes. Anxiety rises for the reduced battery capacity or drive range but the increased safety issues at the low temperature, which dampens the enthusiasm for widespread usage of LIBs. This is largely due to the slowed-down kinetics of Li ions associated with movement and reactions, resulting in increased resistance, reduced power capability, and even dendritic Li plating[1–4].

[1]Beijing National Laboratory for Condensed Matter Physics, Institute of Physics, Chinese Academy of Sciences, Beijing 100190, China. [2]School of Physical Sciences, University of Chinese Academy of Sciences, Beijing 100049, China. [3]College of Materials Science and Opto-Electronic Technology, University of Chinese Academy of Sciences, Beijing 100049, China. [4]Institute of Functional Nano and Soft Materials, Soochow University, Suzhou 215123, China. [5]School of Materials Science and Engineering, University of Shanghai for Science and Technology, Shanghai 200093, China. [6]Xiamen Institute of Rare Earth Materials, Haixi Institutes, Chinese Academy of Sciences, Xiamen 361024, China. [7]State Key Laboratory of Silicon Materials, School of Materials Science and Engineering, Zhejiang University, Hangzhou 310027, China. [8]Tianmu Lake Institute of Advanced Energy Storage Technologies Co. Ltd, Liyang 213300, China. ✉e-mail: liyejing26@gmail.com; zxwang@iphy.ac.cn; wxf@iphy.ac.cn

Thoroughly understanding the influence of temperature on the underlying microstructure and performance of the battery is essential to solving the kinetic bottlenecks and achieving high performance at a low temperature.

Compared with oxides-based cathode materials, the electrochemical performances of the anode materials such as graphite and Li metal are more sensitive to the temperature probably due to the different bulk reaction mechanisms and severe interfacial reaction[5–8]. From the electrolyte to the electrode, Li ions experience 1) solvation in the electrolyte, 2) migration towards the electrode, 3) decomposition to form solid electrolyte interphase (SEI), 4) desolvation, 5) diffusion through the SEI layer, 6) insertion/reduction on the electrode surface, 7) diffusion into the electrode bulk (Fig. 1)[9–11]. These processes occur concurrently or in sequence[12,13]. Among them, desolvation and diffusion through the SEI layer are believed to be the rate-determining steps but the dominant one is still in controversy[14,15]. The activation energy for the desolvation step was measured in a range of 50–80 kJ mol⁻¹ and varies dependent on the electrolyte composition, which contributes to the largest resistance for the charge transfer[16–19]. This value is also affected by the SEI chemistry and changes significantly when different SEI is present[20,21]. Note that separating the desolvation from the Li⁺ diffusion through the SEI layer is always challenging especially when considering the complex structure and limited knowledge of the SEI[22]. Therefore, it is still hard to visualize the above processes, find out the rate-determining steps, and correlate them with the electrochemical performance.

In this sense, herein, the temperature-dependent Li⁺ behavior in Li metal batteries and its relationship with the electrochemical performance were revealed by various characterization tools, including cryogenic high-resolution transmission electron microscopy (cryo-HRTEM), electron energy loss spectroscopy (EELS), and X-ray photoelectron spectroscopy (XPS). The kinetic bottleneck is deciphered as Li⁺ diffusion through the SEI layer at low temperatures. Lowering the temperature not only slows down the kinetics of Li⁺ transport but also changes the thermodynamic reaction of the electrolyte

decomposition, forming an SEI layer consisting of intermediate products rich in organic species, thus increasing the resistance for Li⁺ transport. Tuning the solvation structure of electrolytes with a low LUMO (low unoccupied molecular orbital) energy level and polar groups is beneficial to readily generate an inorganics-rich SEI layer, which has more tolerance to temperature change. These findings help guide the rational design of the interface layer, such as constructing inorganics-rich interphase, to enhance the reaction kinetics of low-temperature Li metal batteries.

## Results

Since the low-temperature performance of Li metal is highly dependent on the electrolyte chemistry[1,23–26], three electrolytes were designed and compared, including 1 mol L⁻¹ lithium hexafluorophosphate (LiPF₆) in ethylene carbonate (EC): dimethyl carbonate (DMC) (1:1, volume ratio) (LiPF₆–EC/DMC, PED), 1 mol L⁻¹ lithium bis(fluorosulfonyl)imide (LiFSI) in EC: DMC (1:1, volume ratio) (LiFSI–EC/DMC, FED) and 1 mol L⁻¹ LiFSI in methyl trifluoroacetate (MTFA): fluoroethylene carbonate (FEC) (8:2, weight ratio) (LiFSI–MTFA/FEC, FMF). The first two electrolytes have the same solvent but different salts (Fig. 2a, b), showing a similar solvation structure as evidenced by the Raman spectra of their electrolytes (Fig. 2d): in both cases Li⁺ is coordinated with EC and DMC, exhibiting characteristic bonds at 729 and 904 cm⁻¹ for EC–Li⁺, and 932 cm⁻¹ for DMC–Li⁺, respectively. Whereas the last two share the same salt but are different in solvents, which results in different solvation structures (Fig. 2b–d). Changes in salt and solvent will alter the SEI structure and component, especially for those having lower LUMO and polar groups, which readily decompose and form inorganics-rich SEI[16,17,27,28]. Based on the calculation, both MTFA (−1.441 eV) and FEC (−0.641 eV) manifest lower LUMO energy levels than EC (−0.602 eV) and DMC (−0.235 eV) (Fig. 2e), suggesting that MTFA and FEC are thermodynamically favorable to decompose and form LiF-rich SEI layer. Such kind of electrolyte and interphase is expected to have enhanced tolerance to the change of operating temperature and thus beneficial for the low-temperature performance of Li metal. With these in mind, a combination of various

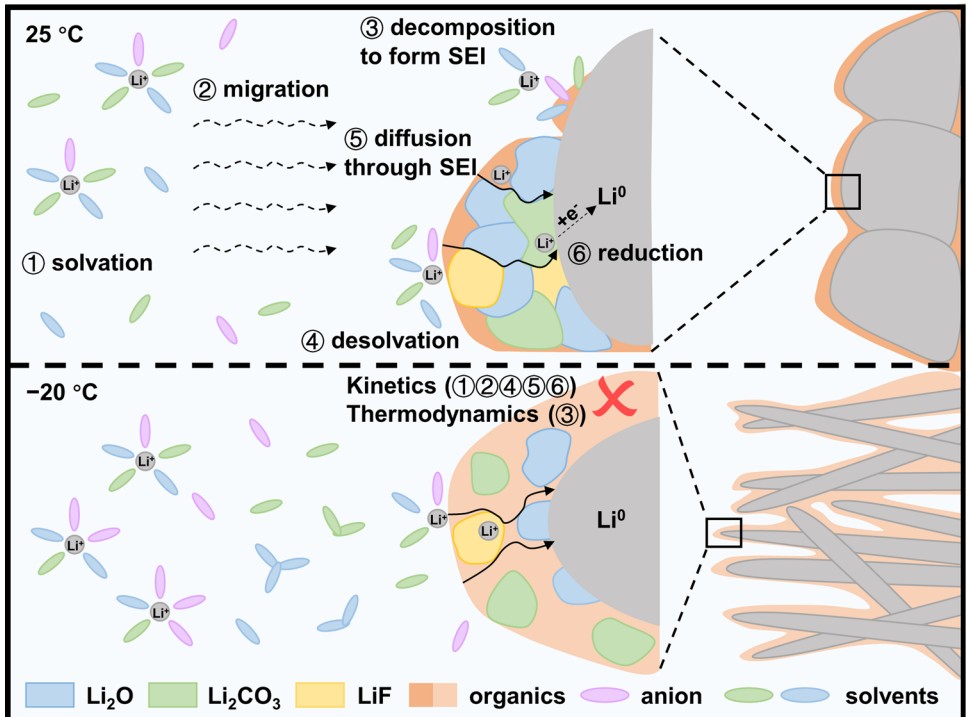

**Fig. 1 | Schematic diagram of ion diffusion and charge transfer during Li plating at room/low temperature.** Lowering the temperature not only slows down Li⁺ transport through the electrolyte and SEI layer, but also leads to incomplete decomposition of the electrolyte, generating SEI layers consisting of metastable intermediate products rich in organics. Therefore, Li metal is prone to grow as dendrites at the low temperature.

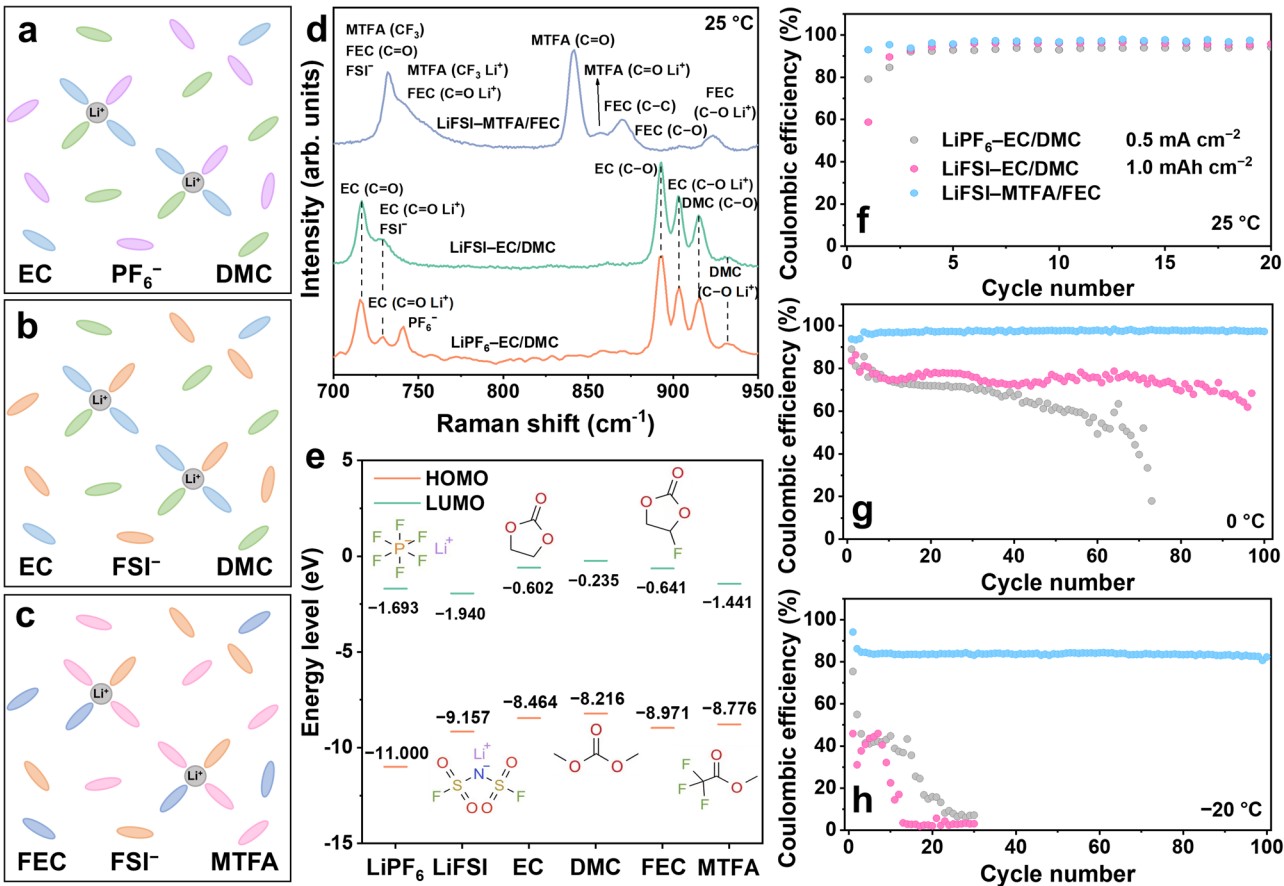

**Fig. 2 | The solvation structures and electrochemical performance. a–c** Schematic illustrations of solvation structure in three electrolytes. **d** Raman spectra of three electrolytes at 25 °C. **e** Molecular structure, HOMO, and LUMO energy levels for LiPF₆, LiFSI, EC, DMC, FEC, and MTFA. Energy levels for LiPF₆, LiFSI, EC, DMC, and FEC are referenced from Wang et al.[60] while that of MTFA was calculated by the same method. **f–h** Coulombic efficiencies of Li‖Cu cells in three electrolytes under a current density of 0.5 mA cm⁻² for 1.0 mAh cm⁻² at different temperatures.

tests and techniques is applied to uncover the temperature effect on the Li metal batteries, probe the changes in SEI chemistry, and reveal the rate-determining steps for the sluggish kinetics at low temperature.

The electrochemical performances were evaluated in the Li‖Cu cells with these three electrolytes at temperatures of 25, 0, and −20 °C (Fig. 2f–h and Supplementary Fig. 1) since they keep liquid (optical photos in Supplementary Fig. 2 and differential scanning calorimetry (DSC) measurement in Supplementary Fig. 3) and maintain the original solvation structure (as evidenced by Raman spectra in Supplementary Fig. 4). Reducing the temperature dramatically decreases the Coulombic efficiencies (CEs, Fig. 2f–h), increases the polarization (Supplementary Fig. 1), and worsens the cycling stability (Fig. 2f–h), especially for those with EC/DMC-based electrolytes. Their CEs are quite low (~84% at 0 °C, ~46% at −20 °C) and drop quickly in the first 10 cycles, indicating the poor reversibility of Li metal and massive accumulation of "dead" Li. The polarization between Li plating and stripping is increased by over 13 times from ~50 mV at 25 °C to ~650 mV at −20 °C, suggestive of the enlarged resistance and the slowed-down kinetics at the low temperature. In comparison, LiFSI−MTFA/FEC displays the highest CE, lowest polarization, and best cycling stability at all temperatures; it shows a stable CE of 83.5% for 100 cycles at −20 °C, which performance is comparable to the state-of-the-art reports (Supplementary Table 1). This suggests that the solvation structure plays a critical role in regulating the electrochemical performance of Li metal by tuning the SEI properties (composition and structure) and desolvation process, especially at low temperatures.

The morphology of the Li deposits was visualized by scanning electron microscopy (SEM). Lowering temperature inhibits the growth

of Li metal and results in deposits with a smaller diameter and tending to grow vertically (Fig. 3). In EC/DMC-based electrolytes, the dendrite-like Li with high tortuosity interweaves at 25 and 0 °C but separately grows into branches of standing pillars at −20 °C due to insufficient ion diffusion and mass transfer. These aggressive Li pillars are apt to penetrate and get stuck in the separator, leaving fewer Li deposits on the current collector. As a consequence, the thickness of residual Li film on the Cu foil is reduced from ~12 μm at 25 °C to ~4 μm at −20 °C in EC/DMC-based electrolyte (Fig. 3 inset) while Li metal stuck in the separator shows content about three times higher than that on Cu foil after initial Li plating for 1.0 mAh cm⁻² (Supplementary Fig. 5) as quantified by titration gas chromatography (TGC). These Li deposits are harmful, not only easily isolated from the current collector and lost electronic connection to form "dead" Li (Supplementary Fig. 6, account for >50% of capacity loss in the EC/DMC-based electrolyte after stripping, indicating that most of the Li⁰ in the separator is irreversible.), but also readily penetrate through the separator and cause an internal short circuit and battery failure[29]. In contrast, albeit the shrunk size, the Li deposits in LiFSI−MTFA/FEC electrolyte are mainly interweaved and remain dense on the Cu foil even at −20 °C, suggesting enhanced ion transport when compared with EC/DMC-based electrolytes.

The above results demonstrate that the electrochemical performances and morphology of Li metal are highly dependent on the temperature (Fig. 1). Lowering the temperature reduces the ion transport and reaction kinetics, resulting in Li deposits with smaller size, longer length towards the separator, and poorer electrochemical reversibility due to the increased porosity and decreased structural

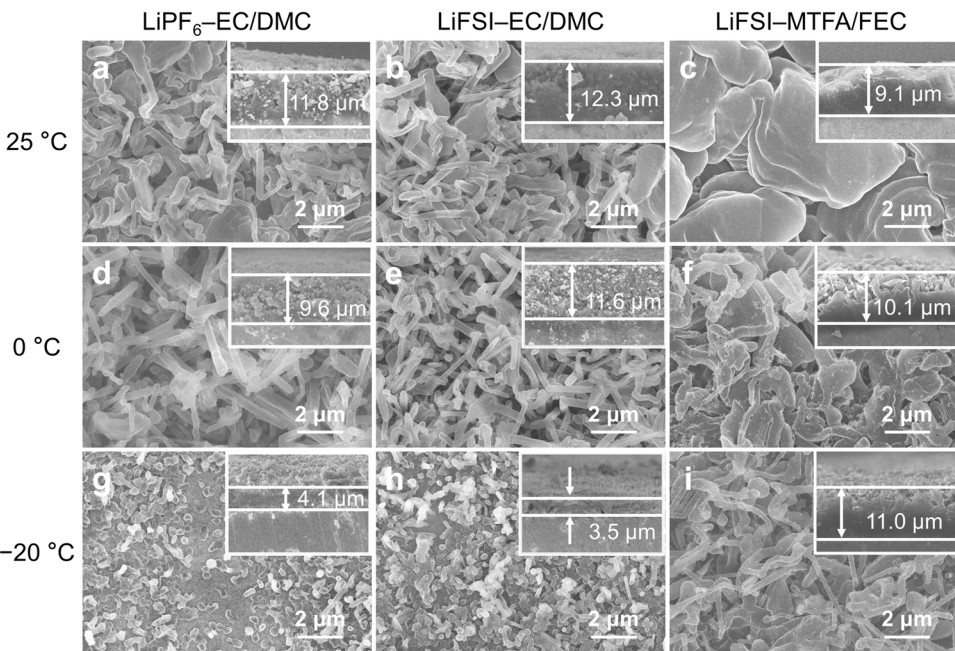

**Fig. 3 | Li metal deposition morphologies.** Top views and cross-section (insets) views of the Li deposits (1.0 mAh cm⁻²) in the Li∥Cu cell using different electrolytes under a current density of 0.5 mA cm⁻² at 25 (**a–c**), 0 (**d–f**) and −20 °C (**g–i**).

connection. Different solvation structure shows varied sensitivity to the temperature, which affects the migration of the solvated Li⁺ in electrolytes, desolvation, and SEI property[9,10,23]. To find the rate-determining step and its relationship with the electrochemical performance at the low temperature, these aspects were further revealed to be dependent on temperature in the following sections.

### Transport of solvated Li ions in the electrolyte

The migration velocity of solvated Li⁺ determines the mass transfer and concentration gradient of Li⁺ before reaching the electrode, which is regulated by the solvation structure and partly reflected in the ionic conductivity. The conductivities of the EC/DMC-based electrolytes are around 11.80, 6.60, and 3.33 mS cm⁻¹ at 25, 0, and −20 °C (Fig. 4a), higher than that of LiFSI−MTFA/FEC electrolyte (5.39, 3.40 and 2.12 mS cm⁻¹). This suggests that although the migration of solvated Li⁺ slows down at low temperatures, it is still higher than the fluorinated electrolyte and not responsible for the poor reversibility of Li metal at −20 °C in EC/DMC-based electrolytes.

### Desolvation process

To discern the main block for the sluggish reaction, the resistance contributions were measured by temperature-dependent electrochemical impedance spectroscopy (EIS) (Fig. 4b–f and Supplementary Figs. 7-10), which can be divided into the ohmic resistance ($R_b$), the SEI layer resistance ($R_{SEI}$), the charge transfer resistance ($R_{ct}$) (Fig. 4c). Compared with the slightly increased ohmic resistance (from -15 to -22 Ω at −20 °C, Fig. 4e), the interfacial resistance ($R_{interface} = R_{ct} + R_{SEI}$) associated with the desolvation process and Li⁺ transporting through the SEI is much higher (ranging from 260 to 1559 Ω at −20 °C, Fig. 4f), implying that the reaction kinetics is dominated by the interfacial reaction.

Since it is not easy to discern the individual contribution of $R_{SEI}$ and $R_{ct}$ from their overlapped spectra in Li∥Cu cells (Supplementary Fig. 7a–c), the distribution of relaxation times (DRT) analysis[30−32] was applied (Supplementary Fig. 7d–f) and a three-electrode cell with $Li_4Ti_5O_{12}$ (LTO) as both working and counter electrode (Fig. 4b) was constructed to minimize the contribution from SEI (Supplementary Fig. 8) and highlight that from charge transfer[12]. The $R_{ct}$ was measured

between two partially-lithiated LTO electrodes (Fig. 4d), and found comparable to that in Li∥Cu cells (Supplementary Figs. 9 and 10a), demonstrating the feasibility of this method and negligible influence of substrates (Supplementary Fig. 10) on the desolvation process when compared with electrolyte. The results show that the $R_{ct}$ increases slowly when the temperature is decreased from 30 to −10 °C and significantly rises to >80 Ω at −20 °C, indicating that the desolvation process is hindered to some extent, especially at a relatively low temperature. The discrepancy in $R_{ct}$ values between different electrolytes is quite small as well as their desolvation energies (inset of Fig. 4d, 33.91 kJ mol⁻¹ for LiPF₆−EC/DMC, 32.88 kJ mol⁻¹ for LiFSI−EC/DMC, and 28.75 kJ mol⁻¹ for LiFSI−MTFA/FEC), which was further proved by density functional theory (DFT) calculation (Supplementary Fig. 11). In contrast, $R_{SEI}$ presents a large difference with different electrolytes, especially at −20 °C, which exhibits 179 Ω for LiFSI−MTFA/FEC while 1148 and 1466 Ω for LiFSI−EC/DMC and LiPF₆−EC/DMC respectively (inset of Fig. 4f). This suggests that it is Li⁺ passing through the SEI layer that significantly alters the reaction kinetics of Li plating, which is highly dependent on the SEI properties.

### SEI Properties

The properties of the SEI layer are regulated by its composition and nanostructure, which was revealed by cryo-HRTEM (Fig. 5 and Supplementary Figs. 12–14), EELS (Fig. 6), atomic force microscopy (AFM, Supplementary Fig. 15), energy-dispersive X-ray spectroscopy (EDS, Supplementary Fig. 16), and XPS (Fig. 7). Electrolyte had been completely removed by rinsing (Supplementary Fig. 17), which excludes its influence on analyzing the SEI.

Cryo-HRTEM images reveal that the SEI layers formed under all conditions are mosaic where crystalline inorganic species (e.g., $Li_2O$, $Li_2CO_3$, and LiF) are embedded in amorphous organic species background (Fig. 5a–i and Supplementary Figs. 12–14). Lowering temperature reduces the concentration of the inorganic species that are unevenly distributed in the SEI layer. Such SEI is believed to have low ionic conductivity and low mechanical strength (Supplementary Fig. 15) to tolerate the large volume change[33,34], thus facilitating the formation of Li dendrites at a low temperature. Consequently, although the side reaction is kinetically alleviated by the low

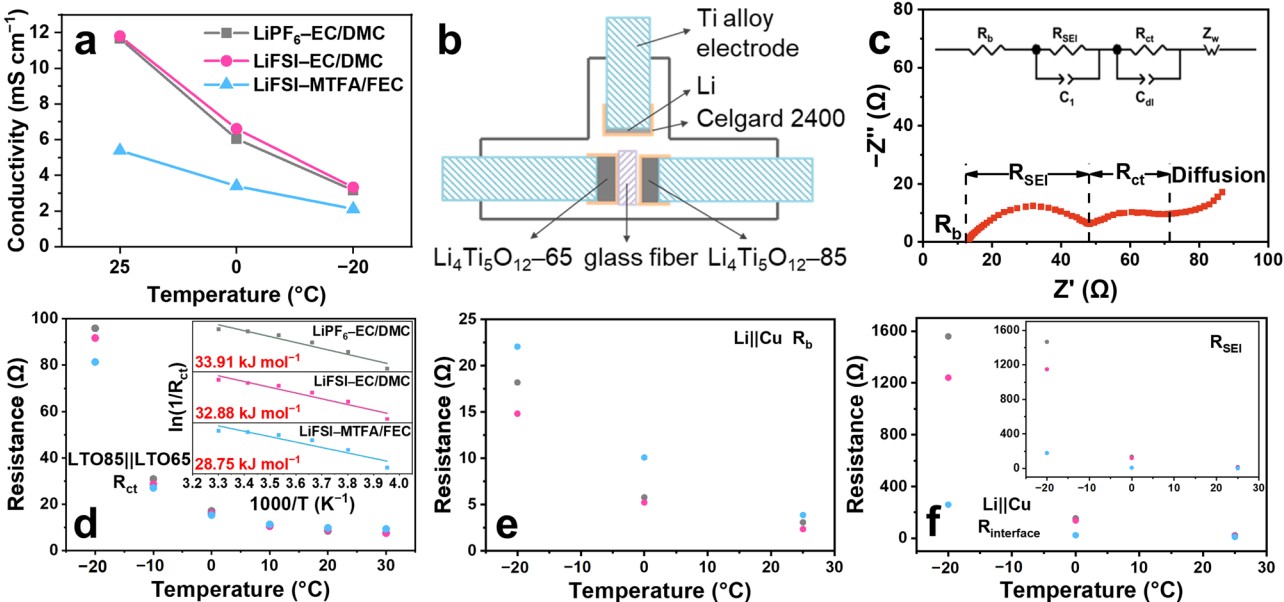

**Fig. 4 | Kinetics of Li deposition. a** Ionic conductivities for the three electrolytes at different temperatures. **b** the setup of the three-electrode cell for the EIS test. **c** Equivalent circuit. **d** The fitting results of $R_{ct}$ in LTO85 ($Li_4Ti_5O_{12}$ electrode partially lithiated to 85 mAh g$^{-1}$)||LTO65 ($Li_4Ti_5O_{12}$ electrode partially lithiated to 65 mAh g$^{-1}$) cell using the equivalent circuit shown in (**c**), the inset of (**d**) is the Arrhenius behavior of the resistance corresponding to Li$^+$ desolvation[12,16,33]. **e,f** The fitting results of $R_b$ and $R_{interface}$ in Li||Cu cell after deposition (0.5 mA cm$^{-2}$, 1.0 mAh cm$^{-2}$), the inset of (**f**) is the $R_{SEI}$ of Li||Cu cell.

temperature resulting in a reduced thickness of SEI on Li deposits (Fig. 5m, based on STEM contrast in three individual random Li deposits as shown in Supplementary Fig. 18), especially in the EC/DMC-based electrolytes, its content (Fig. 5n) in terms of charge transfer determined by TGC is increased dramatically due to the increased surface area of Li deposits. In contrast, the SEI formed in the LiFSI–MTFA/FEC electrolyte shows a rather stable thickness and contents at varied temperatures, potentially a balance of competition among facile decomposition of fluorinated solvents (thermodynamic reaction activity)[35–38], slowed-down reaction kinetics (temperature decreases), and increased reaction sites (surface area increases). The former is evidenced by the more content of SEI formed in the LiFSI–MTFA/FEC than in other electrolytes at room temperature (Fig. 5n), leading to more LiF present in the SEI as indicated by the statistical analysis results (Fig. 5j–l). Such nanostructured SEI is believed beneficial for Li$^+$ transporting across the SEI albeit it is thick, resulting in the lower $R_{SEI}$ in the LiFSI–MTFA/FEC (Fig. 4f).

Besides on the surface of the Li deposits, part of SEI is distributed on the current collector (indirect SEI), which displays nanoballs with a diameter in a range of 50–300 nm. It shrinks and increases in number at the lower temperature, especially in LiFSI–MTFA/FEC (Fig. 6 and Supplementary Figs. 16 and 19). It mainly consists of LiF (F atom % = 33–42%, based on EDS results in Fig. 6d and Supplementary Table 2), $Li_2O$ (O atom% = 4–15%), and organic species (C atom % = 41–55%), where LiF nanograins are surrounded by $Li_2O$ nanograins and amorphous organic species as exhibited by HRTEM (Fig. 6b and Supplementary Fig. 20). This core-shell structure is further confirmed by EELS (Fig. 6c, e) and EDS mapping (Supplementary Fig. 16), in which the core is dominated by LiF (Fig. 6e) and shell thickness is around 18.6 nm (Fig. 6c). This kind of indirect SEI is also present in other electrolytes while its role in regulating Li plating/stripping is under question since it is not intimated to the active materials and may contribute indirectly[34,39]. The heterogeneous structure nature of this indirect SEI is predicated beneficial for Li$^+$ transport, which can homogenize the Li$^+$ flux as an "artificial SEI" on Li metal[40].

Since cryo-TEM is a technique sensitive to the local crystalline species, XPS was carried out as a complementary tool to recognize the SEI composition evolution dependent on temperature, especially for

the organic species (Fig. 7). Similar C 1$s$ and F 1$s$ spectra are shown in the SEI layers of EC/DMC-based electrolytes, suggesting that the SEI composition is closely related to the solvation structure in which organic species such as RC = OLi, ROLi are mainly from the decomposition of solvents while anions contribute to forming LiF and other derivations such as $Li_xPO_yF_z$ from PF$_6^-$ and $SO_2F$ from FSI$^-$ (Fig. 7a, b). These fluorinated species exhibit enhanced signals in the SEI layers of LiFSI–MTFA/FEC due to their additional sources from both MTFA and FEC (Fig. 7b). Decreasing the temperature alters the thermodynamic reaction of electrolyte decomposition, resulting in different reaction pathways and products (Fig. 7c)[41–47]. For example, EC is subjected to ring-opening and C–O breaking to generate RC = OLi, then recombining or decomposition to $Li_2CO_3$/$ROCO_2Li$ and ROLi[42], while the latter reaction is partly hindered at the low temperature. Consequently, $Li_2CO_3$/$ROCO_2Li$ and ROLi are dominated in the SEI formed at 25 and 0 °C but much reduced at −20 °C replaced by its intermediate products RC = OLi as evidenced by the XPS (Fig. 7a, b). In comparison, thanks to the lower LUMO, LiFSI–MTFA/FEC is more thermodynamically favorable to decompose showing more tolerance to the temperature drop and less chemical change of SEI especially when the operating temperature is reduced from 0 to −20 °C (Fig. 7a, b).

## Discussion

The above findings reveal a clear picture of temperature-dependent Li$^+$ behavior in Li metal batteries and its influence on electrochemical performance from both kinetic and thermodynamic aspects (Fig. 1). Lowering the temperature reduces the reaction kinetics, resulting in slowed-down Li$^+$ transport through the electrolyte and SEI layer, and decreased charge transfer for the desolvation process, electrolyte decomposition, and plating. This will lead to increased polarization and growth of Li dendrites. It is worth noting that incomplete decomposition/reaction of the electrolyte (thermodynamics) occurs at low temperatures, forming an SEI layer consisting of intermediate products and rich in organic species. Such SEI is metastable and unfriendly for Li$^+$ transport (Fig. 4f).

Compared with the charge transfer related to desolvation process ($R_{ct}$), Li$^+$ passing through the SEI layer ($R_{SEI}$) is the dominant resistance

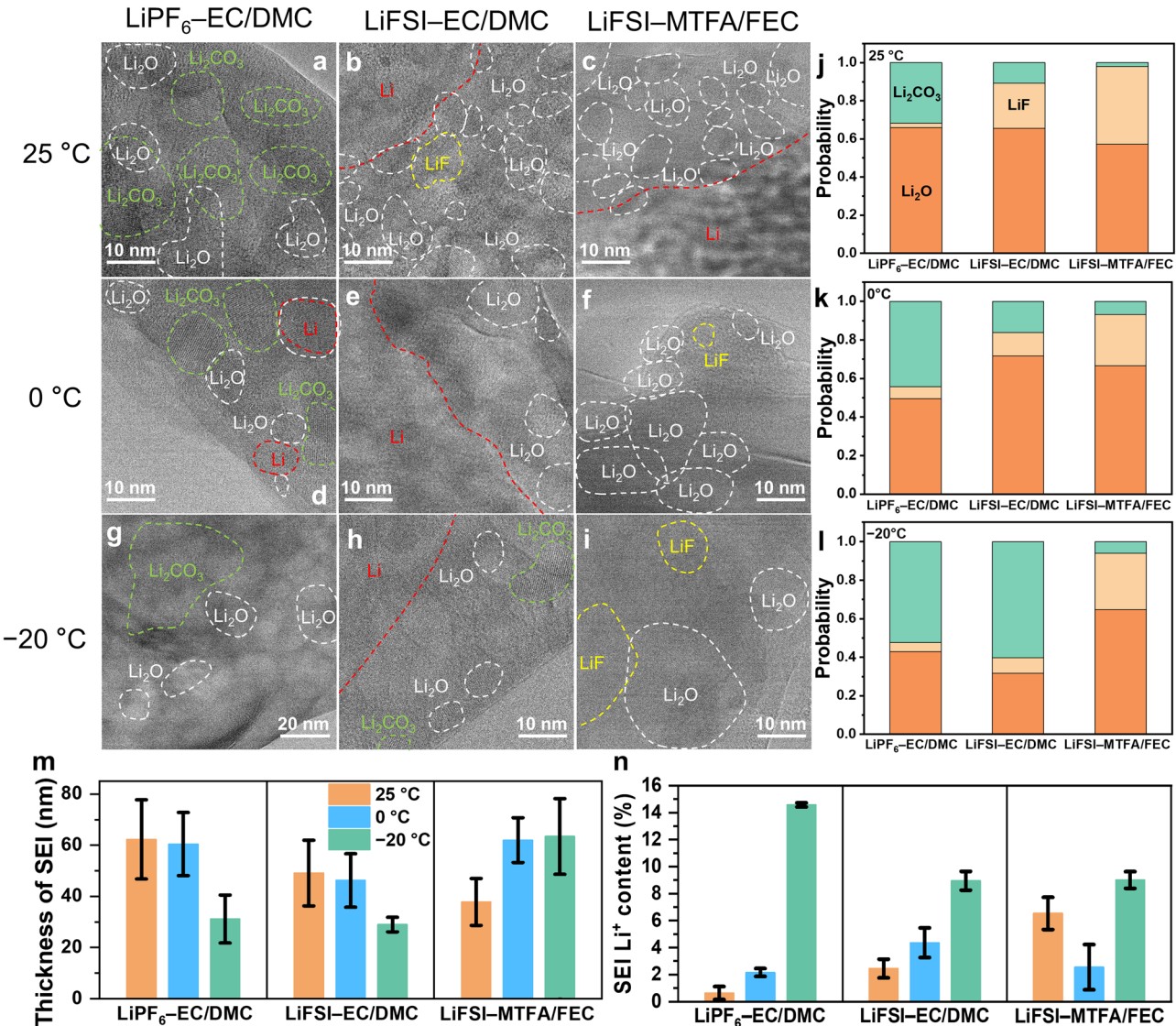

**Fig. 5 | SEI on Li metal.** Cryo-HRTEM images (**a–i**) and statistical analysis (**j–l**) of deposited Li metal using different electrolytes at 25 (**a–c**, and **j**), 0 (**d–f**, and **k**), and −20 °C (**g–i**, and **l**). The thickness (**m**) and content of SEI (**n**) in different electrolytes and temperatures. The error bars in (**m,n**) represent the standard deviation of three independent measurements. (Larger images of (**a–i**) are shown in Supplementary Figs. 12–14).

to the reaction kinetics at the low temperature. This becomes more obvious when the working temperature is lower, such as −40 °C (Supplementary Fig. 21). Huge $R_{SEI}$ value of 2478 Ω is obtained with LiFSI–MTFA/FEC, which is 5.3 times higher than $R_{ct}$ due to the formation of organic-rich SEI layer (Supplementary Fig. 22). Consequently, the rate-determining step for the sluggish kinetics at the low temperature lies in Li⁺ diffusion through the SEI layer, which can be facilitated by the interfacial engineering design of electrolyte and artificial interface.

The composition and nanostructure of the SEI layer are highly dependent on the solvation structure of the electrolyte and its reaction pathways. Salts and solvents, especially with a low LUMO energy level are readily decomposed (Fig. 2e) and those with polar groups contribute to forming more inorganic species in the SEI layer, such as LiF. Such electrolytes and their formed inorganics-rich SEI are found more tolerant to the change of working temperature (thermodynamics) and more beneficial to Li⁺ transport (kinetics)[34,48–50]. This principle has been well demonstrated and proved effective by the fluorinated electrolyte, in which LiF-rich interphase is facile to generate even at low temperature and shows enhanced tolerance to a wide working temperature (from 25 to −70 °C, Supplementary

Fig. 23). Thus, the fluorinated SEI layer enables Li metal anode with high reversibility, low polarization, and high cycling stability at all temperatures. Note that practical application also requires electrolyte work for cathode materials, especially for high-voltage oxides. In this regard, LiFSI–MTFA/FEC fails to work with NCM811 since it starts to decompose at ~3.7 V (Supplementary Fig. 24). Therefore, the Cu‖LiFePO₄ (LFP) pouch cells (250 mAh) with LiFSI–MTFA/FEC electrolyte was assembled and cycled at different temperatures under a specific current of 0.1 C (1 C = 150 mA g⁻¹, Supplementary Fig. 25). They exhibited a specific capacity of 117.8 mAh g⁻¹ during the first 10 cycles at 0 °C (Supplementary Fig. 25a), and its capacity gradually decays as the temperature decreases due to the reduced Li⁺ transport kinetics in the bulk LFP and interphase at low temperatures[51]. Besides electrolyte optimization, a desired SEI can be pre-formed by tuning the operating conditions, such as pre-cycled at room temperature and then changed to the low temperature, which can also improve the low-temperature performance of Li metal in both pouch cells (Supplementary Fig. 25c) and coin cells (Supplementary Fig. 26).

Combining comprehensive characterization techniques, we decoupled the rate-determining step of the Li⁺ transport and charge

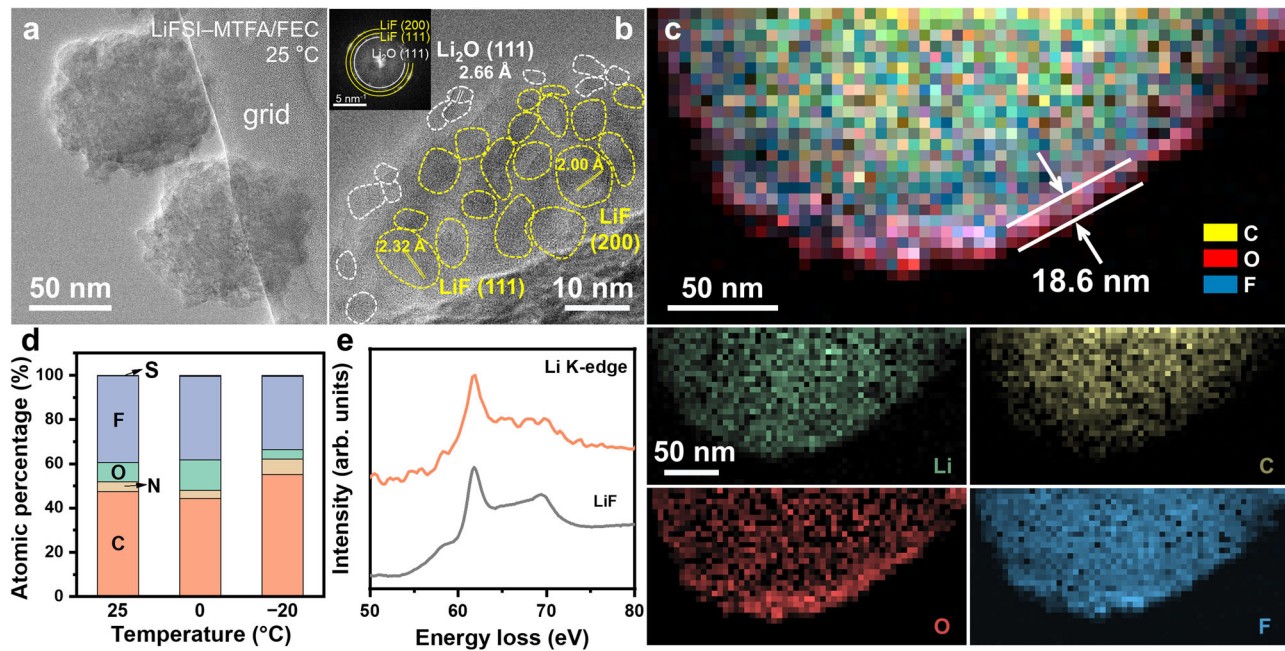

**Fig. 6 | Indirect SEI on the current collector.** Cryo-TEM images (**a,b**), EELS mapping (**c**), EDS results (**d**), and EELS spectra of Li K-edge (**e**) of indirect SEI in LiFSI–MTFA/FEC electrolyte. The inset of (**b**) is the corresponding fast Fourier transform pattern. (Larger image of (**b**) is shown in Supplementary Fig. 20).

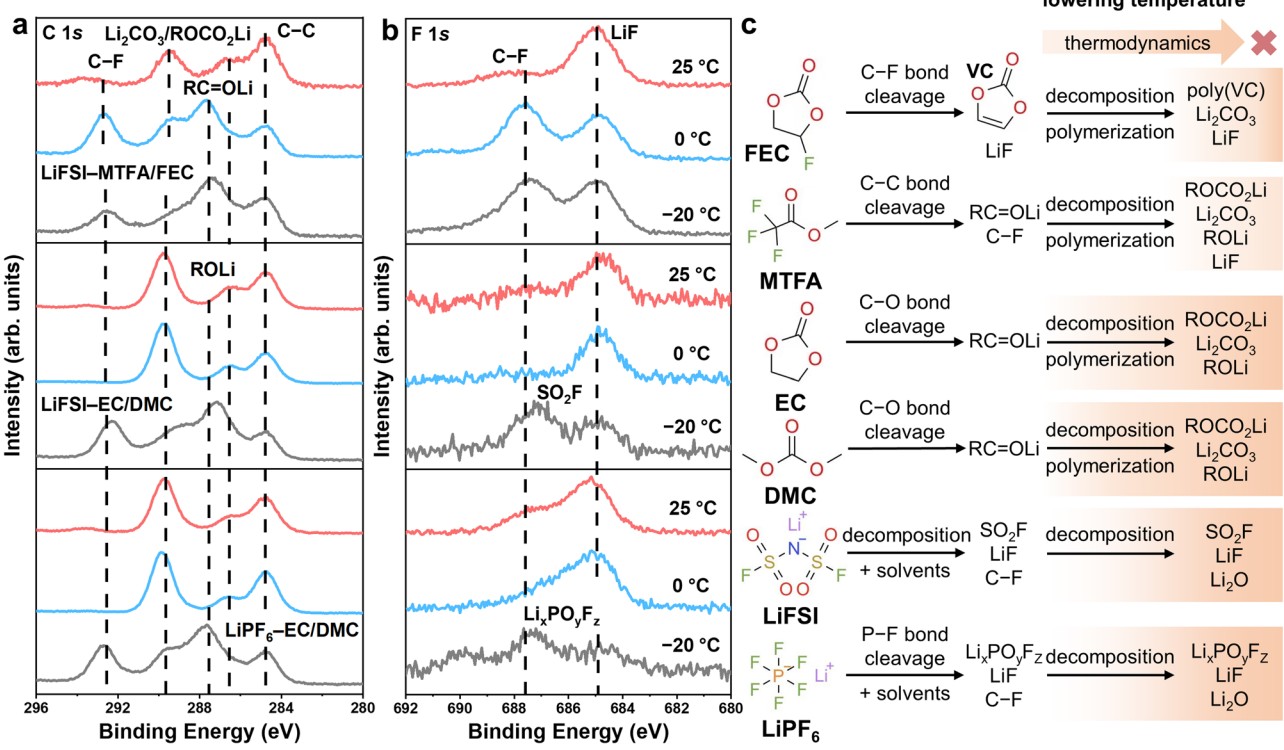

**Fig. 7 | SEI components and reaction pathways. a, b** C 1$s$ (**a**) and F 1$s$ (**b**) XPS spectra of the SEI layer on the deposited Li metal in different electrolytes and temperatures. **c** Bond breaking modes at different temperatures for Li salt and solvents in the three electrolytes.

transfer in Li metal batteries at low temperatures, suggesting that Li$^+$ diffusion through the SEI layer is the bottleneck of the kinetics barrier. Lowering temperature causes incomplete decomposition/reaction of the solvents and salts (thermodynamics), generating SEI layers consisting of intermediate products rich in organic species, thus increasing the resistance for Li$^+$ passing through (kinetics). It is found that fluorinated interphase is beneficial to reducing the above barrier and

enhancing the electrochemical performance of Li metal batteries. These findings reveal the temperature-dependent changes of Li$^+$ behavior and interphase during Li plating/stripping, renew the understanding of the kinetic bottleneck for Li$^+$ transport, and provide the right principles for electrolyte design and interfacial engineering to achieve inorganics-rich interphase and thus high-performance Li metal batteries at the low temperature.

# Methods

## Materials preparation

Battery-grade $LiPF_6$, LiFSI, EC, DMC, FEC and MTFA were ordered from DodoChem Technology. These reagents were used as received without purification. The $LiPF_6$–EC/DMC [1 mol $L^{-1}$ $LiPF_6$ in EC: DMC (1:1, volume ratio)], LiFSI–EC/DMC [1 mol $L^{-1}$ LiFSI in EC: DMC (1:1, volume ratio)] and LiFSI–MTFA/FEC [1 mol $L^{-1}$ LiFSI in MTFA: FEC (8:2, weight ratio)] were prepared in an argon-filled glove box (MBraun Lab Master 130; $O_2 < 0.1$ ppm and $H_2O < 0.1$ ppm). Copper foils (with a diameter of 14 mm) as the lithium deposition substrate were washed with citric acid and alcohol 3 times and then dried up in a vacuum oven at 120 °C for 6 h. The LTO powder was ordered from BTR New Material Group Co., Ltd. without purification. The LTO electrode sheet was prepared by mixing LTO powder, carbon black and polyvinylidene fluoride dissolved in N-methyl pyrrolidone at a weight ratio of 8:1:1 and then casting the slurry onto a piece of copper foil. After vacuum drying at 120 °C for 6 h, The electrode sheet was punched with a diameter of 10 mm and areal loading of ~2.5 mg $cm^{-2}$. LTO65 and LTO85 were prepared by electrochemically lithiated $Li_4Ti_5O_{12}$ electrodes to 65 and 85 mAh $g^{-1}$ at 0.2 C (1 C = 175 mA $g^{-1}$), respectively.

## Electrochemical evaluation

The Li||Cu coin cells (CR2032) were assembled in an argon-filled glove box ($O_2 < 0.1$ ppm and $H_2O < 0.1$ ppm), with copper foil (φ14 mm, 20 μm in thickness) as the working electrode and lithium foil (φ16 mm, 600 μm in thickness) as the counter electrode, glass fiber (φ16.2 mm) as the separator (an additional Celgard 2400 separator (φ16.2 mm) was added between the copper foil and glass fiber in the cells prepared for characterization), and 120 μL of $LiPF_6$–EC/DMC, LiFSI–EC/DMC or LiFSI–MTFA/FEC as the electrolyte. The cells were plating/stripping Li at a current density of 0.5 mA $cm^{-2}$, with a Li deposition capacity of 1.0 mAh $cm^{-2}$ and a charging cut-off voltage of 1 V, unless otherwise specified. Cu||LFP pouch cells with 250 mAh were purchased from LI-FUN Technology Co., Ltd. The electrolyte utilization in pouch cells was 3 g $(Ah)^{-1}$. The voltage range is 3–3.65 V at 0 to −20 °C, and slightly expanded to 3–3.75 V at −30 and −40 °C due to the increased polarization. The electrochemical cycling was performed on a Neware battery test system (CT-4008T-5V10mA-164; Shenzhen). The low-temperature test was carried out in a low-temperature oven (MT3065) produced by Guangzhou-GWS Environmental Equipment Co., Ltd. The copper foils with Li deposits were taken out from the cells, rinsed with DMC, and dried in the vacuum mini-chamber of the glove box before the post-mortem characterization.

The EIS was conducted on an electrochemical workstation (Bio-Logic SP-200 system, France) in the frequency range from 1 MHz to 100 mHz with an a.c. signal of 5 mV. Temperature-dependent EIS was performed using a three-electrode cell with LTO as both working and counter electrode (Fig. 4b) was constructed to minimize the contribution from SEI. The values of $R_b$, $R_{SEI}$ and $R_{ct}$ are fitting using the equivalent circuit shown in Fig. 4c. the $R_{SEI}$(Li||Cu) was calculated by $R_{interface}$(Li||Cu)-$R_{ct}$(LTO85||LTO65).

The activation energy barrier ($E_{ct}$) of desolvation can be obtained by fitting the $R_{ct}$ values based on the Arrhenius equation[12,16,18,19,33,52]:

$$\frac{1}{R_{ct}} = A_0 \, e^{-\frac{E_{ct}}{RT}} \tag{1}$$

where $R_{ct}$, $A_0$, $E_{ct}$, R and T stand for the charge transfer resistance, the pre-exponential constant, the activation energy of desolvation, the standard gas constant and the absolute temperature, respectively.

## Characterization

The Raman spectra of the electrolytes were recorded on a LabRAM HR Evolution Raman spectrometer (532 nm radiation for $LiPF_6$–EC/ DMC and LiFSI–EC/DMC electrolytes while 785 nm radiation for LiFSI–MTFA/FEC to improve the signal to noise ratio; HORIBA) with a resolution of 2 $cm^{-1}$. The morphology of the Li deposits was characterized by SEM (HITACHI S-4800) equipped with a specifically designed sealed container. DSC (NETZSCH STA 449 F3) measurements were carried out from 15 to −100 °C in a sealed aluminum pan with electrolyte, which was cooled down to −100 °C with liquid nitrogen and then scanned from −100 to 15 °C at a rate of 10 °C $min^{-1}$. The freezing point was obtained by taking the onset melting temperature of the endothermic change from the thermal baseline (Supplementary Fig. 3). The XPS was implemented on a Thermo Scientific ESCALAB 250 Xi with monochromatic 150 W (Al Kα line) radiation using a sealed vessel transferred from the glove box to the vacuum chamber. The peak positions were calibrated with the C – C bond (284.8 eV) as reference. The Young's modulus of the SEI layer was obtained by PeakForce QNM mode (Bruker Multimode 8) with the RTESPA-525 tip in an atomic force microscope. FTIR was recorded in attenuated total reflectance mode with a diamond crystal on a Bruker ALPHA II instrument in an argon-filled glove box.

Cryogenic-(scanning) transmission electron microscopy [cryo-(S) TEM] characterizations were carried out using a JEOL JEM-F200 microscope under cryogenic temperatures (−180 °C) at 200 kV. The samples for cryo-(S)TEM characterizations were prepared by directly depositing Li metal in a TEM grid at a current density of 0.5 mA $cm^{-2}$ for 0.5 h. The grid was rinsed by DMC slightly twice and dried in the vacuum mini-chamber of the glove box. Then it was loaded on the cryo-TEM holder (Fischione 2550) equipped with a tip retraction device in the glove box and transferred into the JEOL JEM-F200 microscope without any air exposure with the help of a sealing sleeve (find the schematic illustration of the sample preparation and transfer processes in the reference[53]). Liquid nitrogen was added to the cryo-TEM holder and the sample temperature dropped and stabilized at −180 °C. To minimize the beam damage to the sample, we usually started with an ultralow dose, then gradually increased the dose to improve the signal-to-noise ratio of the image without damaging the sample. The electron beam dose rate is lower than 500 e $Å^{-2}$ $s^{-1}$ and the acquisition time is 0.5 s.

TGC was performed on a Shimadzu Nexis GC-2030 gas chromatography (GC) system equipped with a barrier ionization discharge (BID) detector. The cells after deposition were disassembled in an argon-filled glove box ($O_2 < 0.5$ ppm and $H_2O < 0.5$ ppm). The copper foil with Li deposits (1.0 mAh $cm^{-2}$) together with Celgard 2400 separator was carefully separated from the cell (the glass fiber was separated to measure the metallic Li in the separator in Supplementary Figs. 5–6) and then placed in a sealed container with an open-top cap without washing. To determine the amount of metallic Li[29,54], 0.5 mL of deionized water was injected into the container to react with metallic Li to produce $H_2$ gas. After complete reaction (no visible bubble), the amount of $H_2$ gas was measured by the GC system. The corresponding metallic Li amount was determined according to a pre-established standard calibration curve. To ensure the reliability of the data, three cells were tested in parallel to each condition, and each cell was tested three times.

## Theoretical simulation

The DFT implanted in Gaussian09 software was used to perform the quantum chemistry calculations. The equilibrium state structures with geometry optimization were performed by employing the three-parameter empirical formulation B3LYP in conjunction with the basis set of 6−31 + G(d, p). Then the energies of the highest occupied molecular orbital (HOMO) and lowest unoccupied molecular orbital (LUMO) were analyzed.

Based on gas phase calculations and implicit solvent models using Gaussian16 software package, the desolvation energies of $LiPF_6$–EC/

DMC, LiFSI–EC/DMC and LiFSI–MTFA/FEC electrolytes were studied. The desolvation energies were calculated with the equation:

$$E_{dsv} = E_{Li-solvents/anion} - (E_{Li} + E_{solvents/anion}) \tag{2}$$

where $E_{Li}$, $E_{solvent/anion}$ and $E_{Li\text{-}solvent/anion}$ are the Gibbs free energies of the free Li$^+$, free solvent and anion, and complex, respectively[55].

The structural optimizations of Li-solvents/anion complexes were calculated using the PBE0 level of DFT[56], together with the DEF2TZVP basis sets[57]. The vdW interactions were described using Grimme's dispersion correction[58]. Subsequently, the structures were further optimized using the SMD implicit solvent model[59] with the dielectric constant obtained from molecular dynamics (MD) simulations. All MD simulations were conducted with the GROMACS 2020 program. The systems were firstly equilibrated within the NPT ensemble. During 2 ns NPT simulations, the temperature was controlled at 300 K using a V-rescale thermostat with a relaxation time of 0.1 ps and the pressure was controlled at 1 bar using a Berendsen barostat with a relaxation time of 0.5 ps. Thereafter, the systems were simulated in the NVT ensemble for 1 ns. During 1 ns NVT simulations, the temperature was controlled at 300 K using a V-rescale thermostat with a relaxation time of 0.1 ps.

## Data availability

The data that support the findings of this study are available from the corresponding author upon request.

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

## Acknowledgements

This work was supported by the National Natural Science Foundation of China (NSFC Nos. 22005334 and 52172257, X.W.), the Natural Science Foundation of Beijing (Grant No. Z200013, X.W.), and the National Key Research and Development Program of China (Grant No. 2022YFB2502200, X.W.). The authors also thank Beijing Clean Energy Frontier Research Center of Institute of Physics of Chinese Academy of Sciences and Institutional Center for Shared Technologies and Facilities of Institute of Process Engineering of Chinese Academy of Sciences for necessary characterizations and analysis.

## Author contributions

S.W., X.W., Z.W. and Y.L. conceived the idea and designed the project. X.Z. conducted TGC test. G.Y. performed XPS measurements. B.M. and Q.L. conducted DFT calculations. S.W. performed all the other data collection and analysis. S.Z., Y.L., C.P., H.C., H.Y., X.F., T.C. and L.C. contributed to discussions and interpretation of results. S.W., X.W., Z.W. and Y.L. co-wrote the manuscript, with input from all authors.

## Competing interests

The authors declare no competing interest.
