## [Peer Review File · Nature Communications]

REVIEWER COMMENTS

Reviewer #1 (Remarks to the Author):

The paper deals with SEI-related studies on Li anodes to understand the rate-determining processes at low temperatures in different electrolytes, which in turn determine the composition of the SEI. Since Li⁺ transport across the SEI is known to be rate-limiting at low temperatures, the thermodynamic reactivity of electrolyte species with Li will determine if the SEI is rich in inorganic species or organic species. It is shown here that by altering the solvation structure of electrolyte with a lower LUMO energy level and polar groups, e.g., LiFSI-MTFA/FEC, an inorganic-rich SEI layer is formed which is more tolerant to lower temperatures and more beneficial to Li⁺ transportation. By combining theoretical analysis and experimental studies with a suite of techniques, it is established that fluorinated interphase is beneficial to reducing the thermodynamic and kinetic hindrance at low temperature and enhancing the performance of Li metal batteries. The efficacy of fluorinated salts, e.g., TFFI, FEC, LiPF₆ in enriching the SEI is well known. However, this study extends it lower temperatures and provides a new direction for the development of low temperature electrolytes with low LUMO solvents with fluorinated groups, the same mechanisms that are known to operate at ambient temperature. I recommend publication of this article with minor revisions addressing the comments below:

1) Why is the SEI formation at low temperatures relevant? Is the SEI formed at room temperature during formation prior to switching to low temperatures is completely damaged or the SEI formation on fresh Li formed during cycling is being discussed here?

2) Page 4: "Lowering the temperature not only slows down the kinetic of Li⁺ transportation but also changes the thermodynamic reaction of the electrolyte decomposition, forming an SEI layer consisting of intermediate products rich in organic species".

Why is that at low temperatures the SEI is rich in organics. In other words, why does Li react more readily with solvents instead of salts at low temperature? Don't we expect the reaction to slow down in both cases at low temperature?

3) Page 4: "both Li⁺ is coordinated with EC and DMC" may be changed as "in both cases Li⁺ is coordinated with EC and DMC"

4) Page 5: Why is the CE (coulombic efficiency) that low at low temperatures? Energy efficiency could be low due to polarization losses. But CE should be unless there are reasons the plated Li can't be stripped

(dead Li, SEI formation should be slower at low temperatures). The formation of separator-penetrating pillars due to sluggish mass transfer is not too convincing.

5) In any event, EC/DMC is not the appropriate electrolyte to operate or study at -20°C , possibly close to its freezing point.

6) The use of titanate as WE is not appropriate; Because of its high +ve potential (vs Li), the electrolyte reduction processes and their kinetics will be different and hence the SEI and its effects could be different.

7) From EIS data, we can obtain the SEI resistance and the charge transfer resistance. Diffusion seems to be setting in at lower frequencies. Has that been analyzed and that should correspond to the Li^+ transport across the SEI, correct?

8) With a thicker SEI at room temperature, one would expect a higher SEI resistance at low temperature in MTFA solutions, but that is not what the data at -20°C shows?

9) Are the desolvation energies calculated as a function of temperature?

Reviewer #2 (Remarks to the Author):

In this paper, the authors investigate limiting factors at temperatures down to -20°C when using lithium electrodes in a variety of electrolyte systems. The authors find that transport through the SEI is the limiting factor in these electrolytes at low temperatures, and they characterize the SEI with a few techniques, including XPS and cryo TEM.

The paper is in a timely area, as low-temperature batteries are important and have received increasing attention in recent years. The most novel part of this paper is the direct measurement of charge transfer resistance with the LTO three-electrode cell. Some other aspects of the paper are similar to other work that has been published in the last few years. Overall the paper could be acceptable for publication after addressing the following items:

-The content in figure 1 and 2 (temperature dependent cycling, CE, and lithium morphology) has been discussed in quite a few recent papers in the last 4 years. It would be useful for the authors to present this data in the context of other work that has been done.

-regarding the data in fig 4, I again think this is a creative approach to delineate charge transfer resistance from SEI resistance. The measurement is necessarily done with LTO instead of Li to avoid SEI – but, is there evidence that charge transfer into lithium vs LTO is the same process and has the same resistance value?

-Rct and R_SEI are intrinsically linked, as discussed in the paper. Do the authors have thoughts or arguments on how the presence of SEI may influence Rct, and how this impacts the results in the paper?

-The TEM images in Fig 5 are unfortunately not very useful, since they are too small to see and very little detail is available. It is suggested that one larger image is shown, with remaining large images shown in SI.

-Same issue for TEM image in Fig 6 – very small and poor resolution

Reviewer #3 (Remarks to the Author):

Wang et al. proposed a new fluorinated electrolyte for improving the cycling stability of Li metal anode at $-20\text{ }^{\circ}\text{C}$. Combining electrochemical tests with a series of characterization techniques, they demonstrate that Li⁺ behavior during plating is highly temperature dependent, and designate Li⁺ diffusion in the solid electrolyte interface (SEI) layer as the rate-determining step. However, more rational experimental design, analysis, and more strong results are needed to support this view.

1. In the introductory section, the authors mention that graphite and Li metal are more sensitive at low temperatures than cathodes. However, the fast-reacting lithium deposition process should have better kinetics at low temperature than the intercalation-reacted graphite due to the very different electrode bulk properties. Based on this, the temperature of $-20\text{ }^{\circ}\text{C}$ investigated in this work is not low for Li metal anode, many electrolytes with sufficient liquid range can allow Li metal anode to operate normally at this condition.

2. The schematic at low temperature compared to room temperature is in the form of Li dendrites, which should be discussed here.

3. In Figure 4d, the authors claims that the activation energies of thres electrolytes are similar, in fact, the activation energies than differ by more than 5 KJ mol⁻¹ should not be ignored. In previous report, the difference in activation energy between wealy solvation electrolytes and EC-based electrolytes is also only about 5 KJ mol⁻¹ (Angewandte Chemie International Edition, 2021, 60(8): 4090-4097). In addition, fluorinated carboxylates have also been reported as weak solvating solvents to lower the desolvation energy barrier (Angewandte Chemie International Edition, 2022, 61(36): e202208345, Chem. Commun. 2020, 56, 9640-9643). To this end, molecular dynamics simulations and DFT calculations need to be employed to quantify the desolvation energy of MTFA based electrolyte.

4. If the simulation results also support that the desolvation energy barriers of the three electrolytes are similar, then based on the similar electrolyte ionic conductivity and similar desolvation energy at -20°C, the Li metal with the same SEI film in different electrolytes environments should have similar Coulomb efficiency or cycle performance, even considering the SEI rupture/repair caused by the volume change of Li, the similar performance should be exhibited at least in the initial several cycles. Hence, measurements of SEI formation in LiFSI-MTFA/FEC electrolyte followed by low-temperature cycling in different electrolytes should be supplemented. For this part of the experiment, the author can refer to this paper: ACS applied materials & interfaces, 2017, 9(49): 42761-42768.

5. The relaxation time (DRT) method decouples the intertwined electrochemical steps by capturing the time characteristics of the EIS (Advanced Materials, 2022: 2206448), which can discriminate the individual contributions of RSEI and Rct in Figure 4f.

6. The reaction route/mechanism in Fig. 7c needs to be supported by references.

7. Based on our previous studies, LMBs with fluorinated carboxylate electrolytes have significant overcharge. Do you consider the influence of electrolytes on cathodes, e.g., NCM811? The low temperature performance of NCM811/Li cell are required.

8. The conclusion of this work that RSEI is the rate-determining step of Li metal anode at low temperature is consistent with a recently published graphite-based LIBs, but both at not very low temperatures (-15°C or -20°C). Whether the boundary temperature of RSEI as a rate-determining step is about -20°C, and when the temperature belows -20°C, desolvation determines the low temperature performance?

9. There are some misuses of some symbols in the manuscript. The most obvious is “-”, “-” is the correct minus sign.

Point-by-point Response to the Reviewers' Comments

Nature Communications manuscript NCOMMS-22-40389-T

Title: Temperature-dependent interphase and Li⁺ transportation in lithium metal battery

We sincerely appreciate the reviewer's endorsement of our work, and their insightful and constructive input. The manuscript has now been carefully revised, and the revisions are shown either in **red** for addition or in ~~strike through~~ for deletions.

Note: The figures and tables denoted as Fig. X, Table X; Supplementary Fig. X, Supplementary Table X or Fig. RX, Table RX are in the revised Manuscript, revised Supplementary Information, and Responses to Reviewers, respectively.

Reviewer #1:

The paper deals with SEI-related studies on Li anodes to understand the rate-determining processes at low temperatures in different electrolytes, which in turn determine the composition of the SEI. Since Li⁺ transport across the SEI is known to be rate-limiting at low temperatures, the thermodynamic reactivity of electrolyte species with Li will determine if the SEI is rich in inorganic species or organic species. It is shown here that by altering the solvation structure of electrolyte with a lower LUMO energy level and polar groups, e.g., LiFSI-MTFA/FEC, an inorganic-rich SEI layer is formed which is more tolerant to lower temperatures and more beneficial to Li⁺ transportation. By combining theoretical analysis and experimental studies with a suite of techniques, it is established that fluorinated interphase is beneficial to reducing the thermodynamic and kinetic hindrance at low temperature and enhancing the performance of Li metal batteries. The efficacy of fluorinated salts, e.g., TFFI, FEC, LiPF₆ in enriching the SEI is well known. However, this study extends it lower temperatures and provides a new direction for the development of low temperature electrolytes with low LUMO solvents with fluorinated groups, the same mechanisms

that are known to operate at ambient temperature. I recommend publication of this article with minor revisions addressing the comments below:

Response: We sincerely thank the reviewer for the positive comments on our work.

1. Why is the SEI formation at low temperatures relevant? Is the SEI formed at room temperature during formation prior to switching to low temperatures is completely damaged or the SEI formation on fresh Li formed during cycling is being discussed here?

Response: Thanks for your constructive suggestion. The kinetic and thermodynamic of electrolyte decomposition is closely dependent on the temperature. Lowering temperature will not only slow down the above processes but also stop the reaction before the complete decomposition, forming SEI layer consisting of intermediate products and rich in organic species as evidenced by Cryo-TEM (Fig. 5) and XPS (Fig. 7) results.

According to our results, the pre-formed SEI layer at room temperature will be gradually damaged or replaced by a fresh SEI layer on the Li metal during cycling, and thus having a reducing effect on the Li deposition at the low temperature. As shown in Fig. R1, the cells with the pre-formed SEI at 25 °C for 10 cycles display higher coulombic efficiencies (CE) than those directly cycled at -20 °C especially within the first 20 cycles, demonstrating the positive effect of the SEI pre-formed at 25 °C. However, this difference become smaller during cycling, indicating that the pre-formed SEI is gradually damaged or replaced by the newly-formed SEI at -20 °C. Therefore, these results provide further proof that the SEI property regulates the low-temperature performance of Li metal anode.

Fig. R1. The Coulombic efficiencies of Li||Cu cells with/without pre-formed SEI layer at 25 °C for 10 cycles in three electrolytes under a current density of 0.5 mA cm^{-2} for 1.0 mAh cm^{-2} at $-20 \text{ }^\circ\text{C}$.

[Revisions]

Fig. R1 was added to the revised supplementary information as Supplementary Fig. 23. The following description was added into the revised manuscript: Besides electrolyte optimization, a desired SEI can be pre-formed by tuning the operating conditions, such as pre-cycled at room temperature and then change to the low temperature, which can also improve the low-temperature performance of Li metal (Supplementary Fig. 23).

2. Page 4: “Lowering the temperature not only slows down the kinetic of Li^+ transportation but also changes the thermodynamic reaction of the electrolyte decomposition, forming an SEI layer consisting of intermediate products rich in organic species”

Why is that at low temperatures the SEI is rich in organics. In other words, why does Li react more readily with solvents instead of salts at low temperature? Don't we expect the reaction to slow down in both cases at low temperature?

Response: Thank you for your question. Yes, both reactions of Li with solvents and

lithium salt are slowed down at low temperatures. We did not expect that the Li reacts more readily with solvents than salts at low temperature since the solvation structure of the electrolytes has inconspicuous changes at the temperature range of 25 °C to −20 °C as evidenced by Raman spectra (Fig. R2). The increased content of organics in the SEI is resulted from the incomplete reaction of solvent decomposition at the low temperature (Fig. 7c). For example, the EC/DMC-based electrolyte is first decomposed to RC=OLi, then further decomposed to Li₂CO₃/ROCO₂Li and ROLi; the reaction is stopped at the first step at the low temperature, thus forming SEI consisting of its intermediate products rich in organics as evidenced by the XPS (Fig. 7a-b and Fig. R3) and cryo-TEM (Fig. 5). Therefore, lowering temperature leads to the incomplete reaction of electrolyte, resulting in different reaction pathways and products.

Fig. R2. Raman spectra of LiPF₆-EC/DMC (a), LiFSI-EC/DMC (b) and LiFSI-MTFA/FEC (c) electrolytes at 25 °C, 0 °C and −20 °C.

Fig. R3. (a-b) C 1s (a) and F 1s (b) XPS spectra of the SEI layer on the deposited Li metal in LiPF₆-EC/DMC electrolyte at different temperatures. (c) Bond breaking modes at different temperatures for Li salt and solvents in LiPF₆-EC/DMC electrolyte.

[Revisions]

Fig. R2 was added to the revised supplementary information as Supplementary Fig. 4. The following description was added into the revised manuscript: The electrochemical performances were evaluated in the Li||Cu cells with these three electrolytes at temperatures of 25, 0, and -20 °C (Fig. 2f-h and Supplementary Fig. 1) **since they keep liquid (optical photos in Supplementary Fig. 2 and differential scanning calorimetry (DSC) measurement in Supplementary Fig. 3) and maintain the original solvation structure (as evidenced by Raman spectra in Supplementary Fig. 4).**

3. Page 4: “both Li⁺ is coordinated with EC and DMC” may be changed as “in both cases Li⁺ is coordinated with EC and DMC”.

Response: Thanks for your comment. We checked the whole manuscript carefully and replaced “both Li⁺ is coordinated with EC and DMC” with “in both cases Li⁺ is coordinated with EC and DMC” in the revised manuscript

4. Page 5: Why is the CE (coulombic efficiency) that low at low temperatures? Energy efficiency could be low due to polarization losses. But CE should be unless there are reasons the plated Li can't be stripped (dead Li, SEI formation should slower at low temperatures). The formation of separator-penetrating pillars due to sluggish mass transfer is not too convincing.

Response: Thank you for your question, which improved the quality of our work. The low CE at low temperature is due to the massive formation of SEI layer and “dead” Li^0 , both of which were qualified by titration gas chromatography (TGC). As shown in Fig. 5n and Fig. R4, the SEI Li^+ content in EC/DMC-based electrolytes is dramatically increased owing to the increased surface area of Li deposits albeit the reduced reaction kinetic.

In addition, the Li deposits grow separately into branches of standing pillars and were stuck in the separator at $-20\text{ }^\circ\text{C}$ in EC/DMC-based electrolyte (Fig. 3). The Li^0 content in the separator was found about three times higher than that on Cu foil by TGC after initial Li plating (Supplementary Fig. 5, 80% metallic Li^0 in the separator). After stripping, the “dead” Li^0 content accounts for 54% of capacity loss (Fig. R5), indicating that most of the Li^0 metal in the separator is irreversible.

These results verify that the low CE at low temperature is caused by the massive formation of SEI layer and “dead” Li^0 stuck in the separator.

Fig. R4. SEI content in LiPF_6 -EC/DMC electrolyte at different temperatures. The error bars represent standard deviation.

Fig. R5. The “dead” Li⁰ content of cells in terms of capacity loss with three electrolytes after the initial plating/stripping cycle at $-20\text{ }^{\circ}\text{C}$. The error bars represent standard deviation.

[Revisions]

We added Fig. R5 in the revised supplementary information as Supplementary Fig. 6 and added the corresponding description in the revised manuscript.

These Li deposits are harmful, not only easily isolated from the current collector and lost electronic connection to form “dead” Li (Supplementary Fig. 6, account for $> 50\%$ of capacity loss in the EC/DMC-based electrolyte after stripping, indicating that most of the Li⁰ in the separator is irreversible.), but also readily penetrate through the separator and cause an internal short circuit and battery failure.

5. In any event, EC/DMC is not the appropriate electrolyte to operate or study at $-20\text{ }^{\circ}\text{C}$, possibly close to its freezing point.

Response: Thanks for your comment. Although $-20\text{ }^{\circ}\text{C}$ is close to the freezing point of EC/DMC-based electrolytes, they are still liquid (digital photos in Fig. R6), maintain the almost same solvation structure (as evidenced by Raman spectra in Fig. R2) and have sufficient ionic conductivity for Li⁺ transport ($>3.33\text{ mS cm}^{-1}$ at $-20\text{ }^{\circ}\text{C}$ in Fig. 4a).

Digital photos (Fig. R6) clearly show that the EC/DMC-based electrolytes are still liquid and fluid at $-20\text{ }^{\circ}\text{C}$ but frozen at $-40\text{ }^{\circ}\text{C}$. Differential scanning calorimetry (DSC)

measurements (Fig. R7) suggest that they will be frozen at the temperature lower than $-28\text{ }^{\circ}\text{C}$.

Raman spectra provide information on solvation structures in EC/DMC-based electrolytes at different temperatures (Fig. R2). The solvation structure of electrolyte at $-20\text{ }^{\circ}\text{C}$ is similar to that at 25 and $0\text{ }^{\circ}\text{C}$. In addition, the ionic conductivity of EC/DMC-based electrolytes at $-20\text{ }^{\circ}\text{C}$ was about 3.33 mS cm^{-1} , higher than LiFSI-MTFA/FEC electrolyte (2.12 mS cm^{-1}), which is enough to satisfy the Li^+ conduction in the liquid electrolyte (Fig. 4a). Therefore, in our case, EC/DMC-based electrolytes can operate at $-20\text{ }^{\circ}\text{C}$ and can be used as a reference since it is widely used in the conventional lithium-ion batteries.

Fig. R6. Digital photos of three electrolytes at $-20\text{ }^{\circ}\text{C}$ (a) and $-40\text{ }^{\circ}\text{C}$ (b).

Fig. R7. DSC curves for $\text{LiPF}_6\text{-EC/DMC}$ and LiFSI-EC/DMC electrolytes.

Fig. R2. Raman spectra of LiPF₆-EC/DMC (a), LiFSI-EC/DMC (b) and LiFSI-MTFA/FEC (c) electrolytes at 25 °C, 0 °C and -20 °C.

[Revisions]

We added Fig. R6, R7 and R2 in the revised supplementary information as Supplementary Fig. 2, 3 and 4, respectively. The following description was added into the revised manuscript: The electrochemical performances were evaluated in the Li||Cu cells with these three electrolytes at temperatures of 25, 0, and -20 °C (Fig. 2f-h and Supplementary Fig. 1) since they keep liquid (optical photos in Supplementary Fig. 2 and differential scanning calorimetry (DSC) measurement in Supplementary Fig. 3) and maintain the original solvation structure (as evidenced by Raman spectra in

Supplementary Fig. 4).

In the Method section of revised Manuscript, we add “DSC (NETZSCH STA 449 F3) measurements were carried out from 15 °C to –100 °C in a sealed aluminum pan with electrolyte, which was cooled down to –100 °C with liquid nitrogen and then scanned from –100 °C to 15 °C at a rate of 10 °C min⁻¹. The freezing point was obtained by taking the onset melting temperature of the endothermic change from the thermal baseline (Supplementary Fig. 3).”

6. The use of titanate as WE is not appropriate; Because of its high potential (vs Li), the electrolyte reduction processes and their kinetics will be different and hence the SEI and its effects could be different.

Response: We understand reviewer’s concern and agree that the electrolyte reduction processes and their kinetics may be affected by different electrode materials with varied chemical potential. As noted by reviewer, LTO electrode shows high working potential (~1.5 V vs. Li) and has negligible SEI layer formed on its surface, which is suitable to minimize the contribution of Li⁺ transporting through the SEI and highlight the charge transfer process. Hence, LTO electrode was used in our study to extract the resistance of charge transfer (R_{ct}) during desolvation process and provide reference values for Li metal electrode since they share the similar desolvation processes¹. As shown in Fig. R8, the R_{ct} values of Li||Cu cell are close to those measured between two partially-lithiated LTO electrodes (Fig. R8, for example, 85.14 Ω of Li||Cu and 81.37 Ω of LTO85||LTO65 in FMF), demonstrating the feasibility of this method. The large discrepancy of R_{ct} values between different electrolytes further confirms that the R_{ct} during desolvation process is dominated by the solvation structure rather than the electrode materials.

Meanwhile, the three-electrode setup allows Li⁺ intercalation into the working LTO electrode and the counter LTO electrode, avoiding the potential damage caused during disassembling/re-assembling cells and the influence from the large interfacial resistance from the Li metal if it is used as a WE.

Therefore, LTO electrode is used as a reference as well as an additional complement to

determine the charge transfer resistance R_{ct} during desolvation processes, which results are consistent with that in Li||Cu cell. Compared with R_{SEI} , R_{ct} is much lower at the low temperature (Fig. 4), indicating that Li^+ transporting through the SEI layer is the main rate-limited step.

Fig. R8. The R_{ct} values comparison of LTO85||LTO65 cells and Li||Cu cells.

[Revisions]

We added Fig. R8 in the revised supplementary information as Supplementary Fig. 10a and added the corresponding description in the revised manuscript.

The R_{ct} was measured between two partially-lithiated LTO electrodes (Fig. 4d), and found comparable to that in Li||Cu cells (Supplementary Fig. 8 and 10a), demonstrating the feasibility of this method and negligible influence of substrates (Supplementary Fig. 10) on the desolvation process when compared with electrolyte.

7. From EIS data, we can obtain the SEI resistance and the charge transfer resistance. Diffusion seems to be setting in at lower frequencies. Has that been analyzed and that should correspond to the Li^+ transport across the SEI, correct?

Response: Thank you for your advice. According to the conventional assignments as shown in Fig. 4c, the two semicircles at high frequencies region represent the processes of Li^+ transport across SEI at the higher frequencies and the charge-transfer at the lower

frequencies, whose resistances were usually designated as R_{SEI} and R_{ct} , respectively²⁻⁵. A diagonal line at the lower frequencies region associates with solid diffusion of Li^+ that electroactive species diffuse into or out of the electrodes as a neutral reaction product⁶. Transport of the reacting species is mainly governed by the diffusion obeying the Fick's 1st and 2nd diffusion laws. The diffusion limitation can be expressed as an impedance term in electrochemistry, which is called Warburg impedance (Z_w)⁷. Therefore, we did not analyze the EIS data at the lower frequencies and focused on the two semicircles which indeed reflect the Li^+ transport across the SEI and charge transfer during de-solvation process.

8. With a thicker SEI at room temperature, one would expect a higher SEI resistance at low temperature in MTFAs solutions, but that is not what the data at $-20\text{ }^\circ\text{C}$ shows?

Response: Thank you for your question. We should clarify that the SEI layer resistance (R_{SEI}) is dependent on the thickness and conductivity; the latter is determined by the nanostructure of SEI. Although the SEI is slightly thicker in MTFAs solutions, both cryo-TEM and XPS show that it has more content of inorganic species as well as grain boundaries, which has been proved beneficial to Li^+ transportation^{8,9}. Therefore, albeit the slightly thicker, the lower SEI resistance is thanks to the higher ionic conductivity of SEI formed in the MTFAs solution.

[Revisions]

To clarify this concern, we added following description in the revised manuscript.

Such nanostructured SEI is believed beneficial for Li^+ transporting across the SEI albeit it is thick, resulting in the lower R_{SEI} in the LiFSI-MTFAs/FEC (Fig. 4f).

9. Are the desolvation energies calculated as a function of temperature?

Response: Thank you for your question. Yes, the desolvation energies were calculated as a function of temperature following the approach suggested by Abe et al.^{10,11} and Xu et al.^{1,2}. The activation energies corresponding to charge-transfer processes could be derived from the temperature-dependences of R_{ct} with the assumption that the

transport step is a simple thermally activated process.

$$\frac{1}{R_{ct}} = A_0 e^{-\frac{E_{ct}}{RT}}$$

where R_{ct} , A_0 , E_{ct} , R , and T stand for the charge transfer resistance, the pre-exponential constant, the activation energy of charge transfer, the standard gas constant, and the absolute temperature, respectively.

Both $\text{Li}_4\text{Ti}_5\text{O}_{12}$ (LTO) and Li metal were used as WEs and measured their impedance at a temperature range of $-20\text{ }^\circ\text{C}$ to $30\text{ }^\circ\text{C}$ as shown in Supplementary Fig. 9. The desolvation energies was calculated based on these temperature-dependent R_{ct} and found almost consistence between LTO and Li metal, suggesting the feasibility of this method and that it is dominated by the solvation structure rather than the electrode materials.

[Revisions]

We added the method for calculating the desolvation energies In the Method section of revised manuscript.

Reviewer #2:

In this paper, the authors investigate limiting factors at temperatures down to -20 deg C when using lithium electrodes in a variety of electrolyte systems. The authors find that transport through the SEI is the limiting factor in these electrolytes at low temperatures, and they characterize the SEI with a few techniques, including XPS and cryo TEM.

The paper is in a timely area, as low-temperature batteries are important and have received increasing attention in recent years. The most novel part of this paper is the direct measurement of charge transfer resistance with the LTO three-electrode cell. Some other aspects of the paper are similar to other work that has been published in the last few years. Overall the paper could be acceptable for publication after addressing the following items:

Response: We appreciate the reviewer's positive comments and suggestions which help us improve the quality of our manuscript dramatically.

1. The content in Figure 1 and 2 (temperature dependent cycling, CE, and lithium morphology) has been discussed in quite a few recent papers in the last 4 years. It would be useful for the authors to present this data in the context of other work that has been done.

Response: Thank you for this valuable comment. We admit that the content in Fig. 1 and 2 including temperature dependent cycling, CE, and lithium morphology, has been discussed in quite a few recent papers in the last 4 years and compared the electrochemical performance in the Table R1. Although our electrochemical performance at the low temperature is not among the best, the scope of this manuscript is highlighted on understanding the temperature-dependent Li⁺ behavior during Li plating and finding the rate-determining steps.

Table R1. Comparison of electrochemical performance of Li||Cu (or Li||stainless-steel disk) cells at the low temperature ¹²⁻¹⁸.

Year	Author	Electrolyte	Current density (mA cm ⁻²)	Areal capacity (mAh cm ⁻²)	Cycling life		Initial Coulombic efficiency (%)		Lithium deposits morphology	Reference
					-20 °C	-40 °C	-20 °C	-40 °C		
2019	Wang et al.	1.0 M LiTFSI in DOL/DME (1:1, vol) with 1 wt.% LiNO ₃	1	1	20	-	65.4	-	Roundly shaped (~700 nm at -20 °C) (0.25 mA cm ⁻² -0.15 mAh cm ⁻²)	Nature Energy, 2019, 4(8): 664-670.
2019	Yang et al.	0.3 M LiTFSI + 0.3M THF in FM:CO ₂ (19:1)	0.5	1	-	-	98.6	97.1	-	Joule, 2019, 3(8): 1988-2000.
2019	Thenuwar a et al.	1 M LiTFSI in DOL/DME (8:2, vol)	0.5	0.5	50	40	~88	~85	Short rod-like (~500 nm at -20 °C and ~200 nm at -40 °C) (0.5 mA cm ⁻² -4 mAh cm ⁻²)	Nano Letters, 2019, 19(12): 8664-8672.
		1 M LiTFSI +0.2 M LiNO ₃ in DOL/DME (8:2, vol)	0.5	0.5	50	50	~97	~90	-	
2020	Yang et al.	1.2 M LiTFSI+1 M AN in FM:CO ₂ (19:1)	3	3	-	-	97.1 (-30 °C)	94.6 (-60 °C)	Roundly shaped (2 μm at -60 °C) (0.5 mA cm ⁻² -3 mAh cm ⁻²)	Energy & Environmental Science, 2020, 13(7): 2209-2219.
2020	Thenuwar a et al.	0.8 M LiTFSI+0.2 M LiNO ₃ in DOL/DME (8:2, vol)+10 vol% EC	0.5	0.5	50	40	~88	~85	-	ACS Energy Letters, 2020: 2411-2420.
		0.8 M LiTFSI+0.2 M LiNO ₃ in DOL/DME (8:2, vol)+10 vol% FEC	0.5	0.5	50	50	~97	~90	Roundly shaped (10±2 μm ² at -40 °C) (0.5 mA cm ⁻² -4 mAh cm ⁻²)	
2021	Holoubek et al.	1 M LiFSI DEE	0.5	1	-	-	-	99.0	Massive (~3 μm at -40 °C) (0.5 mA cm ⁻² -5 mAh cm ⁻²)	Nature Energy, 2021, 6(3): 303-313
2022	Cheng et al.	1 M LiFSI+1 wt.% LiNO ₃ in MTHF/THF (6:1, vol)	0.5	1	-	-	~96	~85	Roundly shaped (~300 nm at -40 °C) (1 mA cm ⁻² -1 mAh cm ⁻²)	Advanced functional materials, 2022: 2212349.
2023	This work	1 M LiFSI in MTFA/FEC (8:2, weight)	0.5	1	100	60	94.14	91.10	Dendritic and massive (0.5-2 μm at -20 °C) (0.5 mA cm ⁻² -1 mAh cm ⁻²)	-

[Revisions]

We added Table R1 in the revised supplementary information as Supplementary Table 1 and added the corresponding description in the revised manuscript.

In comparison, LiFSI-MTFA/FEC displays the highest CE, lowest polarization, and

best cycling stability at all temperatures; it shows a stable CE of 83.5% for 100 cycles at $-20\text{ }^{\circ}\text{C}$, which performance is comparable to the state-of-the-art reports (Supplementary Table 1).

2. Regarding the data in figure 4, I again think this is a creative approach to delineate charge transfer resistance from SEI resistance. The measurement is necessarily done with LTO instead of Li to avoid SEI – but, is there evidence that charge transfer into lithium vs LTO is the same process and has the same resistance value?

Response: Thank you again for your positive comment. We understand your concerns and explain this issue in the Reviewer 1, Question 6. Based on our measurement, as shown in Fig. R8, the R_{ct} values of Li||Cu cell are close to those measured between two partially-lithiated LTO electrodes (for example, $85.14\ \Omega$ of Li||Cu and $81.37\ \Omega$ of LTO85||LTO65 in FMF), demonstrating the feasibility of this method. The large discrepancy of R_{ct} values between different electrolytes further confirms that the R_{ct} during desolvation process is dominated by the solvation structure rather than the electrode materials.

Therefore, LTO electrode is suitable to be used as a reference as well as an additional complement to determine the charge transfer resistance R_{ct} during de-solvation processes, which results are consistent with that in Li||Cu cell. Compared with R_{SEI} , R_{ct} is much lower at the low temperature (Fig. 4), indicating that Li^+ transporting through the SEI layer is the main rate-limited step.

Fig. R8. The R_{ct} values comparison of LTO85||LTO65 cells and Li||Cu cells.

[Revisions]

We added Fig. R8 in the revised supplementary information as Supplementary Fig. 10a and added the corresponding description in the revised manuscript.

The R_{ct} was measured between two partially-lithiated LTO electrodes (Fig. 4d), and found comparable to that in Li||Cu cells (Supplementary Fig. 8 and 10a), demonstrating the feasibility of this method and negligible influence of substrates (Supplementary Fig. 10) on the desolvation process when compared with electrolyte.

3. R_{ct} and R_{SEI} are intrinsically linked, as discussed in the paper. Do the authors have thoughts or arguments on how the presence of SEI may influence R_{ct} , and how this impacts the results in the paper?

Response: we appreciate reviewer's comment and admit that R_{ct} and R_{SEI} are intrinsically linked especially due to the porous nature of SEI layer which allows electrolyte to diffuse inside. Following by Kang Xu's opinion¹⁹, we also believe that some species in the SEI could catalyze the breaking up of Li-solvation sheath, and contribute to the rapid completion of the de-solvation process.

However, the R_{ct} in this work is mainly regulated by the solvation structure rather than the substrates as demonstrated in the comparable results between LTO and Li metal

since they also display different SEI. To further confirm this perspective, we formed different SEIs in the PED, FED or FMF electrolyte for 10 cycles at 0 °C and then measured the R_{ct} in the FMF electrolyte after disassembling, rinsing, and reassembling with the same FMF electrolyte. As shown in Fig. R9, these cells with different SEI layers showed varied R_{SEI} but similar R_{ct} in EIS (Fig. R9b), indicating that the R_{ct} is correlated to the bulk electrolyte rather than to the SEI property. Therefore, the influence of different electrode materials and surface SEI layers on R_{ct} can be ignored in our study which does not impact our conclusion.

Fig. R9. The EIS (a) and corresponding distribution of relaxation times (DRT) analysis (b) of the Li||Cu cells precycled in one electrolyte (PED, FED or FMF) for 10 cycles at 0 °C to form SEI layer and then switched to FMF electrolyte, denoted as PED-FMF, FED-FMF and FMF-FMF, respectively.

[Revisions]

We added Fig. R9 in the revised supplementary information as Supplementary Fig. 10b-c and added the corresponding description in the revised manuscript.

The R_{ct} was measured between two partially-lithiated LTO electrodes (Fig. 4d), and found comparable to that in Li||Cu cells (Supplementary Fig. 8 and 10a), demonstrating the feasibility of this method and negligible influence of substrates (Supplementary Fig. 10) on the desolvation process when compared with electrolyte.

Supplementary Fig. 10. (a) The R_{ct} values comparison of LTO85||LTO65 cells and Li||Cu cells. The EIS (b) and corresponding DRT analysis (c) of the Li||Cu cells precycled in one electrolyte (PED, FED or FMF) for 10 cycles at 0 °C to form SEI layer and then switched to FMF electrolyte, denoted as PED-FMF, FED-FMF and FMF-FMF, respectively. Similar value of R_{ct} is obtained with different substrates including LTO/Li metal, and Li metal with varied pre-formed SEI suggesting the negligible influence of substrates on the desolvation process when compared with electrolyte.

4. The TEM images in Figure. 5 are unfortunately not very useful, since they are too small to see and very little detail is available. It is suggested that one larger image is shown, with remaining large images shown in SI.

Same issue for TEM image in Figure 6 – very small and poor resolution.

Response: Thank you for your suggestion. The TEM images in Fig. 5 and Fig. 6 are mainly used to show the components and their distribution in the SEI layers formed on Li deposits under different conditions. Relatively low-magnification images are preferred and used in the Fig. 5 and 6 in order to provide a representative nanostructure of SEI at a relatively large region, which in turn sacrifices the fine structure information. According to the reviewer's advice, we slightly enlarged these TEM images to make them more explicit and provided the corresponding large images with more details in the revised supplementary information.

[Revisions]

Revised Fig. 5-6 and large images in Supplementary Fig. 12-14 and 18 are shown below.

Fig. 5. Characterization of the SEI on Li metal. Cryo-HRTEM images (a-i) and statistical analysis (j-l) of deposited Li metal using different electrolytes at 25 °C (a-c, and j), 0 °C (d-f, and k), and -20 °C (g-i, and l). The thickness (m) and content of SEI (n) in different electrolytes and temperatures. The error bars in m-n represent standard deviation. (Larger images of a-i shown in Supplementary Fig. 12-14)

Fig. 6. Characterization of the indirect SEI on the current collector. Cryo-TEM images (a-b), EELS mapping (c), EDS results (d), and EELS spectra (e) of Li K-edge of indirect SEI in LiFSI-MTFA/FEC electrolyte at 25 °C. (Larger image of b shown in Supplementary Fig. 18)

Supplementary Fig. 12. Cryo-HRTEM images (a-c) and corresponding fast Fourier transform (FFT) pattern (d-f) of deposited Li metal using LiPF₆-EC/DMC electrolyte at different temperatures. (Larger images corresponding to Fig. 5a, d, and g)

Supplementary Fig. 13. Cryo-HRTEM images (a-c) and corresponding FFT pattern (d-f) of deposited Li metal using LiFSI-EC/DMC electrolyte at different temperatures.

(Larger images corresponding to Fig. 5b, e and h)

Supplementary Fig. 14. Cryo-HRTEM images (a-c) and corresponding FFT pattern (d-f) of deposited Li metal using LiFSI-MTFA/FEC electrolyte at different temperatures. (Larger images corresponding to Fig. 5c, f and i)

Supplementary Fig. 18. Cryo-HRTEM image and corresponding FFT pattern of indirect SEI in LiFSI-MTFA/FEC electrolyte at 25 °C. (A larger image corresponding to Fig. 6b)

Reviewer #3:

Wang et al. proposed a new fluorinated electrolyte for improving the cycling stability of Li metal anode at -20 °C. Combing electrochemical tests with a series of

characterization techniques, they demonstrate that Li^+ behavior during plating is highly temperature dependent, and designate Li^+ diffusion in the solid electrolyte interface (SEI) layer as the rate-determining step. However, more rational experimental design, analysis, and more strong results are needed to support this view.

Response: We would like to express our thanks to the reviewer for taking time to review our work, and providing constructive remarks.

1. In the introductory section, the authors mention that graphite and Li metal are more sensitive at low temperatures than cathodes. However, the fast-reacting lithium deposition process should have better kinetics at low temperature than the intercalation-reacted graphite due to the very different electrode bulk properties. Based on this, the temperature of $-20\text{ }^\circ\text{C}$ investigated in this work is not low for Li metal anode, many electrolytes with sufficient liquid range can allow Li metal anode to operate normally at this condition.

Response: Thank you for the reviewer's comment and suggestions. The low temperature of $-20\text{ }^\circ\text{C}$ was selected since all the three electrolytes used in this work maintain liquid (Fig. R6) and their original solvation structure (Fig. R2), which allow the cells to operate normally at $-20\text{ }^\circ\text{C}$. According to DSC results (Fig. R7), EC/DMC-based electrolyte will be frozen at a temperature lower than $-28\text{ }^\circ\text{C}$. Therefore, three typical working temperature of 25, 0, and $-20\text{ }^\circ\text{C}$ is used to study the temperature-dependent interphase and compare the Li^+ behavior in Li metal batteries.

We understand the reviewer's concerns. Following the reviewer's advice, $\text{Li}|\text{Cu}$ cells were tested at $-40\text{ }^\circ\text{C}$ for lithium plating/stripping performance in three electrolytes. As shown in Fig. R10, the LiFSI-MTFA/FEC cells displayed a stable Coulombic efficiency of 82% for 60 cycles at $-40\text{ }^\circ\text{C}$, while those with EC/DMC-based electrolytes could not work due to their solidified electrolyte. Further analysis by EIS and XPS suggests that the Li^+ diffusing through the SEI is still the rate-determining step at $-40\text{ }^\circ\text{C}$.

Fig. R2. Raman spectra of LiPF₆-EC/DMC (a), LiFSI-EC/DMC (b) and LiFSI-MTFA/FEC (c) electrolytes at 25 °C, 0 °C, and -20 °C.

Fig. R6. Digital photos of three electrolytes at -20 °C (a) and -40 °C (b).

Fig. R7. DSC curves for LiPF₆-EC/DMC and LiFSI-EC/DMC electrolytes.

Fig. R10. Coulombic efficiencies of Li||Cu cells in three electrolytes under a current density of 0.5 mA cm^{-2} for 1.0 mAh cm^{-2} at $-40 \text{ }^\circ\text{C}$.

[Revisions]

We added Fig. R10 in the revised supplementary information as Supplementary Fig. 19.

The following description was added into the revised manuscript: Compared with the charge transfer related to desolvation process (R_{ct}), Li^+ diffusion through the SEI layer (R_{SEI}) is the dominant resistance to the reaction kinetic at the low temperature. This become more obvious when the working temperature is lower, such as $-40 \text{ }^\circ\text{C}$ (Supplementary Fig. 19-21).

2. The schematic at low temperature compared to room temperature is in the form of Li dendrites, which should be discussed here.

Response: We are sorry for the insufficient expressions of schematic in the previous manuscript, which we added in the Discussion part as well as in the figure caption of Fig. 1.

[Revisions]

We modified the discussion in the revised manuscript.

The above findings reveal a clear picture of temperature-dependent Li^+ behavior in Li metal batteries and its influence on electrochemical performance from both kinetic and thermodynamic aspects (Fig. 1). Lowering the temperature reduces the reaction kinetics, resulting in slowed-down Li^+ transportation through the electrolyte and SEI layer, and decreased charge transfer for the desolvation process, electrolyte decomposition, and plating. This will lead to increased polarization and growth of Li dendrites. It is worth noting that incomplete decomposition/reaction of the electrolyte (thermodynamics) occurs at low temperatures, forming an SEI layer consisting of intermediate products and rich in organic species. Such SEI is metastable and unfriendly for Li^+ transportation (Fig. 4f).

We added these sentences in the figure caption of Fig. 1 in the revised manuscript.

Lowering the temperature not only slows down Li^+ transportation through the electrolyte and SEI layer, but also leads to incomplete decomposition of the electrolyte, generating SEI layers consisting of metastable intermediate products rich in organics. Therefore, Li metal is prone to grow as dendrites at the low temperature.

3. In Figure 4d, the authors claims that the activation energies of three electrolytes are similar, in fact, the activation energies than differ by more than 5 KJ mol^{-1} should not be ignored. In previous report, the difference in activation energy between weakly solvation electrolytes and EC-based electrolytes is also only about 5 KJ mol^{-1} (Angewandte Chemie International Edition, 2021, 60(8): 4090-4097). In addition,

fluorinated carboxylates have also been reported as weak solvating solvents to lower the desolvation energy barrier (Angewandte Chemie International Edition, 2022, 61(36): e202208345, Chem. Commun. 2020, 56, 9640-9643). To this end, molecular dynamics simulations and DFT calculations need to be employed to quantify the desolvation energy of MTFA based electrolyte.

Response: Thanks for reviewer's comment and we agree that the activation energies that differ by more than 5 KJ mol⁻¹ should not be ignored. This is also the underlying reason for the R_{ct} discrepancy of ~15 Ω between EC/DMC and MTFA based electrolytes at -20 °C. However, large R_{SEI} difference of 1287 Ω is observed (inset of Fig. 4f), suggesting that it is Li⁺ passing through the SEI layer that significantly alters the reaction kinetic of Li plating, which is highly dependent on the SEI properties.

Following reviewer's suggestion, we calculated the desolvation energy barriers of three electrolytes by theoretical simulation with four typical solvation structures (Li⁺ coordinated with a lithium salt anion and three solvents). The desolvation energies were evaluated as the energy difference: E_{dsv}=E_{Li-solvents/anion}-(E_{Li}+E_{solvents/anion}), where E_{Li}, E_{solvent/anion} and E_{Li-solvent/anion} are the Gibbs free energies of the free Li⁺, free solvent and anion, and complex, respectively²⁰.

The result in Fig. R11 shows that LiFSI-MTFA/FEC electrolyte exhibits lower desolvation energy for all solvation configurations. The difference between LiFSI-MTFA/FEC and LiFSI-EC/DMC electrolytes ranges from 2.35 to 3.73 kJ mol⁻¹, close to that between LiFSI-EC/DMC and LiPF₆-EC/DMC electrolytes (2.94-3.95 kJ mol⁻¹). Given the similar Li behavior and electrochemical performance of LiFSI-EC/DMC and LiPF₆-EC/DMC electrolytes, and largely varied Li behavior between LiFSI-EC/DMC and LiFSI-MTFA/FEC electrolytes, the desolvation energy difference of 2.35 to 3.95 kJ mol⁻¹ is not be the dominant factor to regulate the Li plating at the low temperature.

Fig. R11. Calculated desolvation energies of three electrolytes with different solvation configurations.

[Revisions]

We revised the manuscript as below, and Fig. R11 was added to Supplementary Fig. 11 in revised supplementary information.

The discrepancy in R_{ct} values between different electrolytes is quite small as well as their desolvation energies (inset of Fig. 4d, 33.91 kJ mol⁻¹ for LiPF₆-EC/DMC, 32.88 kJ mol⁻¹ for LiFSI-EC/DMC, and 28.75 kJ mol⁻¹ for LiFSI-MTFA/FEC), which was further proved by density functional theory (DFT) calculation (Supplementary Fig. 11).

We added DFT calculation details in the Method section of revised Manuscript.

Based on gas phase calculations and implicit solvent models using Gaussian16 software package, the desolvation energies of LiPF₆-EC/DMC, LiFSI-EC/DMC and LiFSI-MTFA/FEC were studied. The desolvation energies were calculated with the equation: $E_{dsv} = E_{Li-solvents/anion} - (E_{Li} + E_{solvents/anion})$, where E_{Li} , $E_{solvent/anion}$, and $E_{Li-solvent/anion}$ are the Gibbs free energies of the free Li⁺, free solvent and anion, and complex, respectively²⁰. The structural optimizations of Li-solvents/anion complexes were calculated using the PBE0 level of density functional theory²¹, together with the DEF2TZVP basis sets²². The vdW interactions were described using Grimme's dispersion correction²³. Subsequently, the structures were further optimized using the SMD implicit solvent

model²⁴ with the dielectric constant obtained from molecular dynamics (MD) simulations. All MD simulations were conducted with the GROMACS 2020 program. The systems were firstly equilibrated within the NPT ensemble. During 2 ns NPT simulations, the temperature was controlled at 300 K using a V-rescale thermostat with a relaxation time of 0.1 ps and the pressure was controlled at 1 bar using a Berendsen barostat with a relaxation time of 0.5 ps. Thereafter, the systems were simulated in the NVT ensemble for 1 ns. During 1 ns NVT simulations, the temperature was controlled at 300K using a V-rescale thermostat with a relaxation time of 0.1 ps.

4. If the simulation results also support that the desolvation energy barriers of the three electrolytes are similar, then based on the similar electrolyte ionic conductivity and similar desolvation energy at $-20\text{ }^{\circ}\text{C}$, the Li metal with the same SEI film in different electrolytes environments should have similar Coulomb efficiency or cycle performance, even considering the SEI rupture/repair caused by the volume change of Li, the similar performance should be exhibited at least in the initial several cycles. Hence, measurements of SEI formation in LiFSI-MTFA/FEC electrolyte followed by low-temperature cycling in different electrolytes should be supplemented. For this part of the experiment, the author can refer to this paper: ACS applied materials & interfaces, 2017, 9(49): 42761-42768.

Response: Thanks for your constructive suggestion. We also expect that the Li metal with the same SEI film in different electrolytes environments should have similar Coulomb efficiency or cycle performance. Following the reviewer's suggestion, we pre-formed the SEI layer in Li||Cu cells with LiFSI-MTFA/FEC electrolyte at $-20\text{ }^{\circ}\text{C}$ for 10 cycles, and then transferred it to the new cells with the EC/DMC electrolytes. Unfortunately, the pre-formed SEI in LiFSI-MTFA/FEC is so brittle that is easily subjected to damage during cell disassembling, rinsing, drying, and cell re-assembling. As a result, the positive effect of the pre-formed SEI on the low ($-20\text{ }^{\circ}\text{C}$)-temperature electrochemical performance of EC/DMC electrolytes is not so significant but still outperforms than those without pre-formed SEI, especially in the initial several cycles (Fig. R12).

Alternatively, without changing electrolytes and complex transferring processes, we pre-formed SEI layer at room temperature and then cycle the cell at the low temperature. As shown in Fig. R1, the cells with the pre-formed SEI at 25 °C for 10 cycles display higher coulombic efficiencies (CE) than those directly cycled at -20 °C especially within the first 20 cycles, demonstrating the positive effect of the SEI pre-formed at 25 °C. However, this difference become smaller during cycling, indicating that the pre-formed SEI is gradually damaged or replaced by the newly-formed SEI at -20 °C. Therefore, these results provide further proof that the SEI property regulates the low-temperature performance of Li metal anode.

Fig. R12. Coulombic efficiencies of Li||Cu cells under a current density of 0.5 mA cm^{-2} for 1.0 mAh cm^{-2} at $-20 \text{ }^\circ\text{C}$. The cells were pre-formed SEI layer in LiFSI-MTFA/FEC (FMF) electrolyte and then replaced with $\text{LiPF}_6\text{-EC/DMC}$ (PED), LiFSI-EC/DMC (FED) or LiFSI-MTFA/FEC electrolyte, denoted as FMF-PED, FMF-FED and FMF-FMF, respectively.

Fig. R1. The Coulombic efficiencies of Li||Cu cells with/without pre-formed SEI layer at 25 °C for 10 cycles in three electrolytes under a current density of 0.5 mA cm^{-2} for 1.0 mAh cm^{-2} at $-20 \text{ }^\circ\text{C}$.

5. The relaxation time (DRT) method decouples the intertwined electrochemical steps by capturing the time characteristics of the EIS (Advanced Materials, 2022: 2206448), which can discriminate the individual contributions of R_{SEI} and R_{ct} in Figure 4f.

Response: Thank you for your meaningful suggestion. The DRT analysis was conducted to decouple the individual contribution of R_{SEI} and R_{ct} from their overlapped spectra in Li||Cu cells (Supplementary Fig. 7) by time characteristics. The values of R_{SEI} and R_{ct} can be obtained from peak integration in DRT analysis (Fig. R13 and R14). As shown in Fig. R13, R_{SEI} is the rate-determining step of three electrolytes at different temperatures. Compare with EC/DMC-based electrolytes, LiFSI-MTFA/FEC electrolyte presented smaller R_{SEI} and R_{ct} values at sub-zero temperatures. At $-20 \text{ }^\circ\text{C}$, the R_{SEI} of LiFSI-MTFA/FEC is $318.22 \text{ } \Omega$ while that of EC/DMC-based electrolytes increases to $1522.53 \text{ } \Omega$ (PED) and $1249.10 \text{ } \Omega$ (FED), suggesting a faster ion transport kinetics across the SEI in LiFSI-MTFA/FEC. However, the R_{ct} only increases from $85.14 \text{ } \Omega$ (FMF) to $110.69 \text{ } \Omega$ (PED), which is negligible compared to the large discrepancy in R_{SEI} .

Fig. R13. The DRT analysis of the EIS of Li||Cu cells in Supplementary Fig. 7.

Fig. R14. The R_{SEI} and R_{ct} obtained from the integration of the DRT curves in Fig. R13.

[Revisions]

We revised the manuscript as below, and Fig. R13 and R14 were added to Supplementary Fig. 7d-f and 8 in revised supplementary information, respectively.

Since it is not easy to discern the individual contribution of R_{SEI} and R_{ct} from their overlapped spectra in Li||Cu cells (Supplementary Fig. 7a-c), the distribution of relaxation times (DRT) analysis²⁵⁻²⁷ was applied (Supplementary Fig. 7d-f) and a three-electrode cell with $Li_4Ti_5O_{12}$ (LTO) as both working and counter electrode (Fig. 4b) was constructed to minimize the contribution from SEI (Supplementary Fig. 9) and highlight that from charge transfer¹. The R_{ct} was measured between two partially-lithiated LTO electrodes (Fig. 4d), and found comparable to that in Li||Cu cells (Supplementary Fig. 8 and 10a), demonstrating the feasibility of this method and negligible influence of substrates (Supplementary Fig. 10) on the desolvation process when compared with electrolyte.

6. The reaction route/mechanism in Figure 7c needs to be supported by references.

Response: Thank you for your kind suggestion. The main products and reaction

mechanisms in Fig. 7c were inferred by combining the experiments and simulations from both our results and previous reports.

The references (*ECS Electrochem. Lett.* **3**, A91-A93 (2014); *Adv. Theory Simul.* **5**, 2100612 (2022); *Chem. Phys. Lett.* **568-569**, 1-8 (2013))²⁸⁻³⁰ prove that the organic carbonate solvents (EC and DMC) finally decompose to lithium alkylene carbonates (ROCO₂Li) and intermediate C=O species (RC=OLi) are also observed during the multi-step reaction. Subsequently, lithium alkyl carbonates may decompose to form Li₂CO₃.

The reference (*Chem. Mater.* **28**, 8149-8159 (2016))³¹ supports that FEC reduces to form VC and LiF, subsequently VC decomposes to product polymerized VC [poly(VC)] and Li₂CO₃, etc.

The final decomposition products of LiFSI were LiF, Li₂O, SO and NSO as predicted by Liu et al. (*J. Phys. Chem. Lett.* **12**, 2922-2929 (2021))³². Combining the XPS (Fig. 7) and TEM (Fig. 5) results in our experiments, the existence of C-F and SO₂F species indicates that LiFSI was broken on N-S bond and subsequently it may interact with the solvents during decomposition. According to the references (*Chem. Mater.* **31**, 9977-9983 (2019); *RSC Adv.* **3**, 16359-16364. (2013)) and our experiment, the situation is similar for LiPF₆^{33,34}.

The reduction pathway of MTFA was also confirmed from the XPS and TEM results in our experiments

[Revisions]

Following the reviewer's advice, we cited additional references to support our speculation in Fig. 7c as Refs.43-49 in the revised Manuscript.

Decreasing the temperature alters the thermodynamic reaction of electrolyte decomposition, resulting in different reaction pathways and products (Fig. 7c)⁴³⁻⁴⁹.

Refs.43-49 as below.

43. Seo, D. M., Chalasani, D., Parimalam, B. S., Kadam, R., Nie, M. & Lucht, B. L. Reduction reactions of carbonate solvents for lithium ion batteries. *ECS Electrochem. Lett.* **3**, A91-A93 (2014).

44. Liu, Y. et al. Formation of linear oligomers in solid electrolyte interphase via two-electron reduction of ethylene carbonate. *Adv. Theory Simul.* **5**, 2100612 (2022).
45. Leung, K. Two-electron reduction of ethylene carbonate: a quantum chemistry re-examination of mechanisms. *Chem. Phys. Lett.* **568-569**, 1-8 (2013).
46. Michan, A. L. et al. Fluoroethylene carbonate and vinylene carbonate reduction: understanding lithium-ion battery electrolyte additives and solid electrolyte interphase formation. *Chem. Mater.* **28**, 8149-8159 (2016).
47. Liu, Y. et al. Effects of high and low salt concentrations in electrolytes at lithium-metal anode surfaces using DFT-ReaxFF hybrid molecular dynamics method. *J. Phys. Chem. Lett.* **12**, 2922-2929 (2021).
48. Henschel, J., Peschel, C., Günter, F., Reinhart, G., Winter, M. & Nowak, S. Reaction product analysis of the most active “inactive” material in lithium-ion batteries—the electrolyte. II: battery operation and additive impact. *Chem. Mater.* **31**, 9977-9983 (2019).
49. Wilken, S., Treskow, M., Scheers, J., Johansson, P. & Jacobsson, P. Initial stages of thermal decomposition of LiPF₆-based lithium ion battery electrolytes by detailed Raman and NMR spectroscopy. *RSC Adv.* **3**, 16359-16364. (2013).

7. Based on our previous studies, LMBs with fluorinated carboxylate electrolytes have significant overcharge. Do you consider the influence of electrolytes on cathodes, e.g., NCM811? The low temperature performance of NCM811/Li cell are required.

Response: Thanks for your good suggestion. We did not investigate the influence of the electrolyte on cathode materials since we focused on the Li metal behavior rather than the practical application of the electrolytes. As suggested by reviewer, we evaluated the electrochemical performances of Li||NCM811 cells with three electrolytes at 25 °C (Fig. R15a). Unfortunately, the cells with LiFSI-MTFA/FEC and LiFSI-EC/DMC electrolytes cannot work normally due to the low oxidative voltage of MTFA and LiFSI, which start to decompose at about 3.7 V and 4.0 V (*vs.* Li), respectively. The linear sweep voltammetry (LSV) of LiFSI-MTFA/FEC also proves that it is oxidized from a lower potential (~3.7 V *vs.* Li) (Fig. R15b). Therefore, LiFSI-MTFA/FEC is not an appropriate electrolyte for high-voltage cathode materials.

Fig. R15. (a) The initial charge/discharge profiles of Li||NCM811 cells with three electrolytes with a voltage range of 2.7–4.3 V at 25 °C. (b) LSV profile of LiFSI-MTFA/FEC electrolyte on a Ti alloy electrode at a scan rate of 0.1 mV s⁻¹.

[Revisions]

We added Fig. R15 in the revised supplementary information as Supplementary Fig. 22 and added corresponding description in the revised manuscript.

Note that practical application also requires electrolyte work for cathode materials, especially for high-voltage oxides. In this regard, LiFSI-MTFA/FEC failed to work with NMC811 since it starts to decompose at ~3.7 V (Supplementary Fig. 22).

8. The conclusion of this work that R_{SEI} is the rate-determining step of Li metal anode at low temperature is consistent with a recently published graphite-based LIBs, but both at not very low temperatures (-15°C or -20°C). Whether the boundary temperature of R_{SEI} as a rate-determining step is about -20°C , and when the temperature below -20°C , desolvation determines the low temperature performance?

Response: This is a very good question. R_{SEI} remains the rate-determining step of LiFSI-MTFA/FEC electrolyte when temperature below -20°C . In order to solve the reviewer's concerns, the main block for lithium plating/stripping in LiFSI-MTFA/FEC electrolyte at -40°C was discerned by EIS (Fig. R16a). DRT analysis was used to decouple the individual contribution of R_{SEI} and R_{ct} . The R_{SEI} value is 5.3 times higher than R_{ct} , suggesting that Li^+ passing through the SEI layer remains the rate-determining step at -40°C (Fig. R16b). At the same time, XPS was carried out to recognize the

SEI composition. The intermediate products, such as C-F, RC=OLi, and SO₂F, are dominated in the SEI layer formed at -40 °C and inorganic species like LiF are negligible (Fig. R17). The results verified that decreasing the temperature alters the thermodynamic reaction of electrolyte decomposition, forming an SEI layer consisting of intermediate products and rich in organic species. Such change of SEI layer leads to slower kinetics of Li⁺ transport through SEI layer. Therefore, when the temperature below -20 °C, Li⁺ passing through the SEI layer is still the rate-determining step.

Fig. R16. (a) The Nyquist plot of Li||Cu cell with LiFSI-MTFA/FEC electrolyte at -40 °C after initial deposition (0.5 mA cm⁻², 1.0 mAh cm⁻²). (b) The DRT analysis of EIS in (a).

Fig. R17. C 1s (a) and F 1s (b) XPS spectra of the SEI layer on the deposited Li metal in different electrolytes and temperatures.

[Revisions]

We added Fig. R16 and Fig. R17 as Supplementary Fig. 20 and 21 in revised supplementary information, respectively.

In the Discussion section of revised Manuscript, we added “Huge R_{SEI} value of 2478 Ω is obtained with LiFSI-MTFA/FEC, which is 5.3 times higher than R_{ct} due to the formation of organic-rich SEI layer (Supplementary Fig. 20-21).”

9. There are some misuses of some symbols in the manuscript. The most obvious is “-”, “-” is the correct minus sign.

Response: Thank you very much for pointing out this error. We have checked the whole manuscript carefully and replaced improper “-” with “-” in the main text and figures of the revised manuscript and revised supplementary information.

References for Response

1. Xu, K., von Cresce, A. & Lee, U. Differentiating contributions to “ion transfer” barrier from interphasial resistance and Li^+ desolvation at electrolyte/graphite interface. *Langmuir* **26**, 11538-11543 (2010).
2. Xu, K. “Charge-transfer” process at graphite/electrolyte interface and the solvation sheath structure of Li^+ in nonaqueous electrolytes. *J. Electrochem. Soc.* **154**, A162-A167 (2007).
3. Yao, Y. X. et al. Regulating interfacial chemistry in lithium-ion batteries by a weakly solvating electrolyte. *Angew.Chem. Int. Ed.* **60**, 4090-4097 (2021).
4. Zhang, S. et al. Tackling realistic Li^+ flux for high-energy lithium metal batteries. *Nat. Commun.* **13**, 5431 (2022).
5. Yang, G. et al. Lithium plating and stripping on carbon nanotube sponge. *Nano Lett.* **19**, 494-499 (2019).
6. Huggins, R. A. Simple method to determine electronic and ionic components of the conductivity in mixed conductors a review. *Ionics* **8**, 300-313 (2002).
- 7 Karden, E. Using Low Frequency Impedance Spectroscopy for Characterization,

Monitoring, and Modeling of Industrial Batteries (Shaker, 2002).

8. Yuan, S. et al. Revisiting the designing criteria of advanced solid electrolyte interphase on lithium metal anode under practical condition. *Nano Energy* **83**, 105847 (2021).
9. Sun, C. et al. 50C fast-charge Li-ion batteries using a graphite anode. *Adv. Mater.* **34**, e2206020 (2022).
10. Abe, T., Fukuda, H., Iriyama, Y. & Ogumi, Z. Solvated Li-ion transfer at interface between graphite and electrolyte. *J. Electrochem. Soc.* **151**, A1120-A1123 (2004).
11. Abe, T., Sagane, F., Ohtsuka, M., Iriyama, Y. & Ogumi, Z. Lithium-ion transfer at the interface between lithium-ion conductive ceramic electrolyte and liquid electrolyte—a key to enhancing the rate capability of lithium-ion batteries. *J. Electrochem. Soc.* **152**, A2151-A2154 (2005).
12. Thenuwara, A. C. et al. Efficient low-temperature cycling of lithium metal anodes by tailoring the solid-electrolyte interphase. *ACS Energy Lett.* **5**, 2411-2420 (2020).
13. Thenuwara, A. C., Shetty, P. P. & McDowell, M. T. Distinct nanoscale interphases and morphology of lithium metal electrodes operating at low temperatures. *Nano Lett.* **19**, 8664-8672 (2019).
14. Wang, J. et al. Improving cyclability of Li metal batteries at elevated temperatures and its origin revealed by cryo-electron microscopy. *Nat. Energy* **4**, 664-670 (2019).
15. Yang, Y. et al. High-efficiency lithium-metal anode enabled by liquefied gas electrolytes. *Joule* **3**, 1986-2000 (2019).
16. Yang, Y. et al. Liquefied gas electrolytes for wide-temperature lithium metal batteries. *Energy Environ. Sci.* **13**, 2209-2219 (2020).
17. Holoubek, J. et al. Tailoring electrolyte solvation for Li metal batteries cycled at ultra-low temperature. *Nat. Energy* **6**, 303-313 (2021).
18. Cheng, L. et al. An ultrafast and stable Li-metal battery cycled at -40°C . *Adv. Funct. Mater.*, 2212349 (2022).
19. Xu, K. & von Wald Cresce, A. Li^+ -solvation/desolvation dictates interphasial processes on graphitic anode in Li ion cells. *J. Mater. Res.* **27**, 2327-2341 (2012).
20. Liu, J. et al. Reconstruction of LiF-rich interphases through an anti-freezing

- electrolyte for ultralow-temperature LiCoO₂ batteries. *Energy Environ. Sci.*, doi: 10.1039/D2EE02411J (2023).
21. Grimme, S., Ehrlich, S. & Goerigk, L. Effect of the damping function in dispersion corrected density functional theory. *J. Comput. Chem.* **32**, 1456-1465 (2011).
22. Marenich, A. V., Cramer, C. J. & Truhlar, D. G. Universal solvation model based on solute electron density and on a continuum model of the solvent defined by the bulk dielectric constant and atomic surface tensions. *J. Phys. Chem. B* **113**, 6378-6396 (2009).
23. Cornell, W. D. et al. A second generation force field for the simulation of proteins, nucleic acids, and organic molecules *J. Am. Chem. Soc.* **117**, 5179-5197 (1995).
24. Wang, J., Wolf, R. M., Caldwell, J. W., Kollman, P. A. & Case, D. A. Development and testing of a general amber force field. *J. Comput. Chem.* **25**, 1157-1174 (2004).
25. Schmidt, J. P., Chrobak, T., Ender, M., Illig, J., Klotz, D. & Ivers-Tiffée, E. Studies on LiFePO₄ as cathode material using impedance spectroscopy. *J. Power Sources* **196**, 5342-5348 (2011).
26. Illig, J., Ender, M., Chrobak, T., Schmidt, J. P., Klotz, D. & Ivers-Tiffée, E. Separation of charge transfer and contact resistance in LiFePO₄-cathodes by impedance modeling. *J. Electrochem. Soc.* **159**, A952-A960 (2012).
27. Yao, Y. X. et al. Ethylene-carbonate-free electrolytes for rechargeable Li-ion pouch cells at sub-freezing temperatures. *Adv. Mater.* **34**, e2206448 (2022)
28. Seo, D. M., Chalasani, D., Parimalam, B. S., Kadam, R., Nie, M. & Lucht, B. L. Reduction reactions of carbonate solvents for lithium ion batteries. *ECS Electrochem. Lett.* **3**, A91-A93 (2014).
29. Liu, Y. et al. Formation of linear oligomers in solid electrolyte interphase via two-electron reduction of ethylene carbonate. *Adv. Theory Simul.* **5**, 2100612 (2022).
30. Leung, K. Two-electron reduction of ethylene carbonate: a quantum chemistry re-examination of mechanisms. *Chem. Phys. Lett.* **568-569**, 1-8 (2013).
31. Michan, A. L. et al. Fluoroethylene carbonate and vinylene carbonate reduction: understanding lithium-ion battery electrolyte additives and solid electrolyte interphase formation. *Chem. Mater.* **28**, 8149-8159 (2016).
32. Liu, Y. et al. Effects of high and low salt concentrations in electrolytes at lithium-metal anode surfaces using DFT-ReaxFF hybrid molecular dynamics method. *J. Phys. Chem. Lett.* **12**, 2922-2929 (2021).

33. Henschel, J., Peschel, C., Günter, F., Reinhart, G., Winter, M. & Nowak, S. Reaction product analysis of the most active “inactive” material in lithium-ion batteries—the electrolyte. II: battery operation and additive impact. *Chem. Mater.* **31**, 9977-9983 (2019).
34. Wilken, S., Treskow, M., Scheers, J., Johansson, P. & Jacobsson, P. Initial stages of thermal decomposition of LiPF₆-based lithium ion battery electrolytes by detailed Raman and NMR spectroscopy. *RSC Adv.* **3**, 16359-16364. (2013).

REVIEWER COMMENTS

Reviewer #1 (Remarks to the Author):

The authors have adequately addressed the comments from me and the other two reviewers in the revision. The revised manuscript is now acceptable for publication.

Reviewer #3 (Remarks to the Author):

The authors have tried to response the reviewers' comments and the quality of this work has been improved. However, there are some further points should be well addressed.

1. The incomplete decomposition/reaction of the solvents and salts induces the formation of SEI layers consisting of intermediate products rich in organic species at low temperature. How to decipher the electrolyte/SEI with incomplete decomposition/reaction of the solvents and salts? Some of the incomplete decomposed solvents can be join the electrolyte phase, especially the temperature increases to room temperature.
2. As shown in cycling performance in Figure R1, the SEI is unstable. How about the mechanical properties of SEI formed at room temperature and -20oC? Can you apply AFM probe to touch the SEI? It is widely accepted that fluorinated interphase is beneficial to enhancing the electrochemical performance of Li metal batteries. The mechanical properties should be highly concerned.
3. How about the Columbic efficiency and cycling stability at more harsh condition ranging from -40~-100oC? This fields have been highly concerned by several groups and the temperature dependent interphase and Li⁺ transportation is also a very important issue.
4. The TEM images herein is unclear now. How to judge the edge of Li₂O, Li₂CO₃, and LiF? It is widely known the HRTEM is the potential of the atom in the thin sample and how to avoid the beam damage and physical damage during sample preparation should be further explained.
5. Can you demonstrate the cycling performance of the Li metal cell with 1 mol L⁻¹ LiFSI in methyl trifluoroacetate (MTFA): fluoroethylene carbonate (FEC) (8:2, w/w) at low temperature in pouch cell? The cycling performance in pouch cell is highly expected to full demonstration of the actual match between electrolyte and electrode in practical conditions.

Point-by-point Response to the Reviewers' Comments

Nature Communications manuscript NCOMMS-22-40389A

Title: Temperature-dependent interphase and Li⁺ transportation in lithium metal battery

We sincerely appreciate the reviewer's endorsement of our work, and their insightful and constructive input. The manuscript has now been carefully revised, and the revisions are shown either in **red** for addition or in ~~strike through~~ for deletions.

Note: The figures and tables denoted as Fig. X, Table X; Supplementary Fig. X, Supplementary Table X or Fig. RX, Table RX are in the revised Manuscript, revised Supplementary Information, and Responses to Reviewers, respectively.

Reviewer #1 (Remarks to the Author):

The authors have adequately addressed the comments from me and the other two reviewers in the revision. The revised manuscript is now acceptable for publication.

Response: We sincerely appreciate the reviewer's endorsement of our work.

Reviewer #3 (Remarks to the Author):

The authors have tried to respond the reviewers' comments and the quality of this work has been improved. However, there are some further points should be well addressed.

Response: We appreciate the reviewer's positive comments and suggestions. Following the reviewer's suggestions, we have addressed the comments point by point and revised the manuscript thoroughly.

1. The incomplete decomposition/reaction of the solvents and salts induces the formation of SEI layers consisting of intermediate products rich in organic species at low temperature. How to decipher the electrolyte/SEI with incomplete decomposition/reaction of the solvents and salts? Some of the incomplete decomposed solvents can be join the electrolyte phase, especially the temperature increases to room

temperature.

Response: Thank you for this valuable comment and we understand reviewer's concern that it maybe difficulty to discipher the electrolyte and SEI with incomplete decomposition products if they are coexisting since they may have similar functional groups. To avoid the influence from trace electrolyte, the samples for post-mortem characterizations (*e.g.* SEM, XPS, TEM, *etc.*) were fully rinsed by dimethyl carbonate (DMC) solvent to remove the residual electrolyte and dried in the vacuum mini-chamber of the glove box. If there are some electrolytes left on the electrode, the signals from salts should be apparently detected by TEM and XPS. However, in our experiments, we did not find any signals belonging to salts by TEM (Fig. 5) and XPS (Fig. 7), indicating that the electrolyte is completely removed and the detected singals are actually from SEI layer.

Considering the technical limits of XPS (sensitive to surface) and TEM (sensitive to local area), we conducted Fourier-transformed infrared (FTIR) on the Li deposits in LiFSI–MTFA/FEC (FMF) electrolyte with/without DMC rinsing and compared their spectra with that from electrolyte (Fig. R1). It is obviously to find the signal from the electrolyte in the unrinsed sample but not the rinsed one, suggesting that the electrolyte can be completely removed by rinsing. Only typical SEI components were observed on the rinsed sample, such as Li_2CO_3 (peaks at 886, 1461, and 1525 cm^{-1} for CO_3^{2-}), ROCO_2Li (peaks at 1767 and 1780 cm^{-1} for C=O) and ROLi (peak at 1296 cm^{-1} for C–O) ¹⁻⁴. These results demonstrate that the SEI layers rich in intermediate products can be detected without the influence from the electrolyte after proper rinsing process.

Fig. R1. FTIR spectra of LiFSI–MTFA/FEC electrolyte and deposited Li after/without rinsing.

[Revisions]

We added Fig. R1 in the revised supplementary information as Supplementary Fig. 17 and added the corresponding description in the revised manuscript.

Electrolyte had been completely removed by rinsing (Supplementary Fig. 17), which excludes its influence on analyzing the SEI.

The following description was added into the Methods of revised manuscript:

FTIR was recorded in attenuated total reflectance mode with a diamond crystal on a Bruker ALPHA II instrument in an argon-filled glove box.

2. As shown in cycling performance in Figure R1, the SEI is unstable. How about the mechanical properties of SEI formed at room temperature and $-20\text{ }^{\circ}\text{C}$? Can you apply AFM probe to touch the SEI? It is widely accepted that fluorinated interphase is beneficial to enhancing the electrochemical performance of Li metal batteries. The mechanical properties should be highly concerned.

Response: Thank you for your suggestion. Following reviewer's suggestion, atomic force microscopy (AFM) were conducted to probe the mechanical property of SEI layer formed at $25\text{ }^{\circ}\text{C}$ and $-20\text{ }^{\circ}\text{C}$ (Fig. R2). The FMF-derived SEI layer at $25\text{ }^{\circ}\text{C}$ manifests a significantly higher average Young's modulus (11.1 GPa) than $-20\text{ }^{\circ}\text{C}$ (8.2 GPa), representing an enhanced capability in suppressing the dendrite growth. This is because lowering temperature alters the thermodynamic reaction of electrolyte decomposition, forming SEI layer consisting of intermediate products and rich in organic species, which have poor mechanical properties^{5,6}. Therefore, the cells with the pre-formed SEI at $25\text{ }^{\circ}\text{C}$ for 10 cycles display higher coulombic efficiencies (CE) than those directly cycled at $-20\text{ }^{\circ}\text{C}$ (Supplementary Fig. 26).

Fig. R2. Mechanical property of SEI layer formed at 25 and $-20\text{ }^{\circ}\text{C}$ in LiFSI–MTFA/FEC electrolyte. The FMF-derived SEI layer at $25\text{ }^{\circ}\text{C}$ manifests a significantly higher average Young’s modulus (11.1 GPa) than $-20\text{ }^{\circ}\text{C}$ (8.2 GPa), representing an enhanced capability in suppressing the dendrite growth.

[Revisions]

Fig. R2 was added to the revised supplementary information as Supplementary Fig. 15 and added the corresponding description in the revised manuscript.

Lowering temperature reduces the concentration of the inorganic species that are unevenly distributed in the SEI layer. Such SEI is believed to have low ionic conductivity and low mechanical strength (Supplementary Fig. 15) to tolerate the large volume change^{34, 35}.

The following description was added into the Methods of revised manuscript: The Young’s modulus of SEI layer was obtained by PeakForce QNM mode (Bruker Multimode 8) with the RTESPA-525 tip in an atomic force microscope.

3. How about the Columbic efficiency and cycling stability at more harsh condition ranging from $-40\sim-100\text{ }^{\circ}\text{C}$? This fields have been highly concerned by several groups and the temperature dependent interphase and Li^+ transportation is also a very important issue.

Response: Thanks for your constructive suggestion. Following the reviewer's advice, Li||Cu cells were tested at a harsh condition ranging from $-40 - -70\text{ }^{\circ}\text{C}$ for lithium

plating/stripping in LiFSI–MTFA/FEC electrolyte. $-70\text{ }^{\circ}\text{C}$ is the lowest temperature that our low-temperature oven (MT3065, Guangzhou-GWS Environmental Equipment Co., Ltd) can be achieved. As shown in Fig. R3, the CE degrades significantly with decreasing temperature, from $\sim 82\%$ at $-20\text{ }^{\circ}\text{C}$ to 68.3% at $-60\text{ }^{\circ}\text{C}$, and further to 43.2% at $-70\text{ }^{\circ}\text{C}$, indicating that the LiFSI–MTFA/FEC electrolyte is not suitable for $-70\text{ }^{\circ}\text{C}$ and even lower temperatures.

Fig. R3. Coulombic efficiencies of Li||Cu cells in LiFSI–MTFA/FEC electrolyte under a current density of 0.5 mA cm^{-2} for 1.0 mAh cm^{-2} at $25\text{ }^{\circ}\text{C}$ – $-70\text{ }^{\circ}\text{C}$.

[Revisions]

Fig. R3 was added to the revised supplementary information as Supplementary Fig. 23 and added the corresponding description in the revised manuscript.

This principle has been well demonstrated and proved effective by the fluorinated electrolyte, in which LiF-rich interphase is facile to generate even at low temperature and shows enhanced tolerance to a wide working temperature (from $25\text{ }^{\circ}\text{C}$ to $-70\text{ }^{\circ}\text{C}$, Supplementary Fig. 23).

The following description was added into the Methods of revised manuscript: The low-temperature test was carried out in a low-temperature oven (MT3065) produced by Guangzhou-GWS Environmental Equipment Co., Ltd.

4. The TEM images herein is unclear now. How to judge the edge of Li_2O , Li_2CO_3 , and LiF? It is widely known the HRTEM is the potential of the atom in the thin sample and how to avoid the beam damage and physical damage during sample preparation should be further explained.

Response: Thank you for your advice. We improved the resolution of TEM image in manuscript (Figs. 5-6) and provided the corresponding large and high-resolution image with more details in the supplementary information (Supplementary Figs. 12-14 and 20).

The edge of inorganic grains such as Li_2O , Li_2CO_3 , and LiF can be distinguished by their lattice fringes in the HRTEM and inversed FFT (iFFT) patterns; the latter is more useful and is able to separate the signals from the overlapped grains since its signal is based on the selected characteristic diffraction points/rings of the specific components. Fig. R4 shows the detailed procedures by iFFT method. First, FFT was performed on the HRTEM image (Fig. R4a) to identify phases and corresponding crystal orientation (Fig. R4b). Then a mask was applied to selected diffraction points/rings, such as Li_2CO_3 in Fig. R4c-d. After inversed FFT operation, the corresponding position was obtained in the real space, which was highlighted by False color (Fig. R4e). Based on the False color area, the edge of the inorganic grains could be clearly identified in the HRTEM image (Fig. R4f). The above steps were repeated to label the other crystalline inorganic grains (Fig. R4g).

Fig. R4. Procedures to identify the species and distribution of the crystalline inorganic grains in the SEI by inversed FFT method.

To reduce the beam damage, we imaged the sample with the minimized the dose at the temperature of $-180\text{ }^{\circ}\text{C}$, which have be proved effective in our previous work^{5, 7-11} as well as other reports¹²⁻¹⁷. Cryogenic temperature can reduce reactivity and enhance stability of sample, but it may still melt, shrink, and sublime if high electron beam dose is used. Therefore, experimentally, we usually started with an ultralow dose, then gradually increased the dose to improve the signal-to-noise ratio of the image without damaging the sample. The electron beam dose rate of this work is lower than $500\text{ e } \text{\AA}^{-2}\text{ s}^{-1}$, which is similar to other works^{11-13, 18}.

In order to avoid the physical damage, TEM grids were placed in the cells to collect Li deposits approaching its real electrochemical states. After disassembly, the grid was gently rinsed with DMC to avoid over-washing or mechanical damage. Then it was loaded onto the specially designed TEM holders (Fischione 2550) equipped with a tip retraction device in the argon-filling glove box, and transferred into the microscope without any air exposure with the help of a sealing sleeve. The above procedures had been well developed in our group and its schematic illustration (Fig. R5) was demonstrated in our previous paper⁶.

Fig. R5. Schematic illustration of the sample preparation and transfer processes for cryo-TEM detection of the deposited Na⁶.

[Revisions]

The following description was added into the Methods of revised manuscript: **Then it was loaded on the cryo-TEM holder (Fischione 2550) equipped with a tip retraction device in the glove box and transferred into JEOL JEM-F200 microscope without any air exposure with the help of a sealing sleeve (find the schematic illustration of the sample preparation and transfer processes in the reference⁵⁴).** Liquid nitrogen was added to the cryo-TEM holder and the sample temperature dropped and stabilized at -180°C . **To minimize the beam damage to the sample, we usually started with an ultralow dose, then gradually increased the dose to improve the signal-to-noise ratio of**

the image without damaging the sample. The electron beam dose rate is lower than $500 \text{ e } \text{Å}^{-2} \text{ s}^{-1}$ and the acquisition time is 0.5 s.

5. Can you demonstrate the cycling performance of the Li metal cell with 1 mol L^{-1} LiFSI in methyl trifluoroacetate (MTFA): fluoroethylene carbonate (FEC) (8:2, w/w) at low temperature in pouch cell? The cycling performance in pouch cell is highly expected to full demonstration of the actual match between electrolyte and electrode in practical conditions.

Response: Thank you for this valuable comment. Per reviewer's request, we assembled the Cu||LiFePO₄ (LFP) pouch cells (250 mAh) with LiFSI–MTFA/FEC electrolyte and cycled them at different temperature under a current density of 0.1 C (1 C=150 mA g⁻¹). LFP was used due to the low oxidation potential of the electrolyte (~3.7 V at 25 °C). The voltage range is 3–3.65 V at 0–20 °C, and slightly expanded to 3–3.75 V at –30 and –40 °C due to the increased polarization (Fig. R6). The Cu||LFP pouch cells exhibited a specific capacity of 117.8 mAh g⁻¹ during the first 10 cycles at 0 °C (Fig. R6a). Subsequently, it decays rapidly as the temperature decreases due to the sluggish Li⁺ transportation kinetics in bulk LFP at low temperatures¹⁹. The specific capacity of Cu||LFP pouch cell maintains 52.8 mAh g⁻¹ after 20 cycles at –20 °C (Fig. R6c). With pre-formed SEI layer at room temperature, the cell present slightly higher capacity (57.5 mAh g⁻¹) and longer cycling life (30 cycles), demonstrating that the SEI pre-formed at 25 °C is beneficial for Li⁺ transportation at the low temperature at pouch cells.

Fig. R6. (a) The electrochemical performance and (b) voltage profiles of Cu||LFP pouch cells at different temperatures. (c) The cycling performance of Cu||LFP pouch cells with/without pre-formed SEI layer at 25 °C for 3 cycles in LiFSI–MTFA/FEC electrolyte at –20 °C.

[Revisions]

Fig. R6 was added to the revised supplementary information as Supplementary Fig. 25. The following description was added into the revised manuscript:

Therefore, the Cu||LiFePO₄ (LFP) pouch cells (250 mAh) with LiFSI–MTFA/FEC electrolyte was assembled and cycled at different temperature under a current density of 0.1 C (1 C=150 mA g⁻¹, Supplementary Fig. 25). They exhibited a specific capacity of 117.8 mAh g⁻¹ during the first 10 cycles at 0 °C (Supplementary Fig. 25a), and its capacity gradually decays as the temperature decreases due to the reduced Li⁺ transportation kinetics in the bulk LFP and interphase at low temperatures⁵². Beside electrolyte optimization, a desired SEI can be pre-formed by tuning the operating conditions, such as pre-cycled at room temperature and then change to the low temperature, which can also improve the low-temperature performance of Li metal in both pouch cells (Supplementary Fig. 25c) and coin cells (Supplementary Fig. 26).

The following description was added into the Methods of revised manuscript:

The electrolyte utilization in the Cu||LFP pouch cells was 3 g (Ah)⁻¹. The voltage range is 3–3.65 V at 0––20 °C, and slightly expanded to 3–3.75 V at –30 and –40 °C due to the increased polarization.

References for Response

1. Li, L., et al. Transport and electrochemical properties and spectral features of non-aqueous electrolytes containing LiFSI in linear carbonate solvents. *J. Electrochem. Soc.* **158**, A74-A82 (2011).
2. Ruther, R. E., Hays, K. A., An, S. J., Li, J., Wood, D. L., Nanda, J. Chemical evolution in silicon-graphite composite anodes investigated by vibrational spectroscopy. *ACS Appl. Mater. Interfaces* **10**, 18641-18649 (2018).
3. Yang, G., Li, Y., Liu, S., Zhang, S., Wang, Z., Chen, L. LiFSI to improve lithium deposition in carbonate electrolyte. *Energy Storage Mater.* **24**, 160-166 (2019).
4. Zhang, S. et al. Understanding the dropping of lithium plating potential in carbonate electrolyte. *Nano Energy* **70**, 104486 (2020).
5. Yuan, S. et al. Revisiting the designing criteria of advanced solid electrolyte interphase on lithium metal anode under practical condition. *Nano Energy* **83**, 105847 (2021).
6. Zheng, X. et al. Deciphering the role of fluoroethylene carbonate towards highly reversible sodium metal anodes. *Research* **2022**, 9754612 (2022).
7. Huang, Y. et al. Eco-friendly electrolytes via a robust bond design for high-energy Li metal batteries. *Energy Environ. Sci.* **15**, 4349-4361 (2022).
8. Li, Z. et al. Oxygen-permeable and moisture-proof membrane for stable Li-O₂/air batteries in humid working environment. *Energy Storage Mater.* **58**, 94-100 (2023).
9. Li, Z., Yu, R., Weng, S., Zhang, Q., Wang, X., Guo, X. Tailoring polymer electrolyte ionic conductivity for production of low-temperature operating quasi-solid-state lithium metal batteries. *Nat. Commun.* **14**, 482 (2023).
10. Zhang, K. et al. Chlorinated dual-protective layers as interfacial stabilizer for dendrite-free lithium metal anode. *Energy Storage Mater.* **41**, 485-494 (2021).
11. Wang, X. et al. New insights on the structure of electrochemically deposited lithium metal and its solid electrolyte interphases via cryogenic TEM. *Nano Lett.* **17**, 7606-7612 (2017).
12. Li, Y. et al. Atomic structure of sensitive battery materials and Interfaces revealed by cryo-electron microscopy. *Science* **358**, 506-510 (2017).

13. Li, Y., Huang, W., Li, Y., Pei, A., Boyle, D. T., Cui, Y. Correlating structure and function of battery interphases at atomic resolution using cryoelectron microscopy. *Joule* **2**, 2167-2177 (2018).
14. Fang, C. et al. Quantifying inactive lithium in lithium metal batteries. *Nature* **572**, 511-515 (2019).
15. Zachman, M. J., Tu, Z., Choudhury, S., Archer, L. A., Kourkoutis, L. F. Cryo-STEM mapping of solid-liquid interfaces and dendrites in lithium-metal batteries. *Nature* **560**, 345-349 (2018).
16. Huang, W., Wang, H., Boyle, D. T., Li, Y., Cui, Y. Resolving nanoscopic and mesoscopic heterogeneity of fluorinated species in battery solid-electrolyte interphases by cryogenic electron microscopy. *ACS Energy Lett.* **5**, 1128-1135 (2020).
17. Ju, Z. et al. Biomacromolecules enabled dendrite-free lithium metal battery and its origin revealed by cryo-electron microscopy. *Nat. Commun.* **11**, 488 (2020).
18. Yang, G. et al. Iron carbide allured lithium metal storage in carbon nanotube cavities. *Energy Storage Mater.* **36**, 459-465 (2021).
19. Rui, X. H., Jin, Y., Feng, X.Y., Zhang, L. C., Chen, C. H. A comparative study on the low-temperature performance of LiFePO_4/C and $\text{Li}_3\text{V}_2(\text{PO}_4)_3/\text{C}$ cathodes for lithium-ion batteries. *J. Power Sources* **196**, 2109-2114 (2011).

REVIEWERS' COMMENTS

Reviewer #3 (Remarks to the Author):

The authors have revised the manuscript very carefully. It can be accepted now.